# Global perturbation of organic carbon cycling by river damming

Taylor Maavara[1], Ronny Lauerwald[2,3], Pierre Regnier[2] & Philippe Van Cappellen[1]

The damming of rivers represents one of the most far-reaching human modifications of the flows of water and associated matter from land to sea. Dam reservoirs are hotspots of sediment accumulation, primary productivity ($P$) and carbon mineralization ($R$) along the river continuum. Here we show that for the period 1970–2030, global carbon mineralization in reservoirs exceeds carbon fixation ($P < R$); the global $P/R$ ratio, however, varies significantly, from 0.20 to 0.58 because of the changing age distribution of dams. We further estimate that at the start of the twenty-first century, in-reservoir burial plus mineralization eliminated $4.0 \pm 0.9$ Tmol per year ($48 \pm 11$ Tg C per year) or 13% of total organic carbon (OC) carried by rivers to the oceans. Because of the ongoing boom in dam building, in particular in emerging economies, this value could rise to $6.9 \pm 1.5$ Tmol per year ($83 \pm 18$ Tg C per year) or 19% by 2030.

[1] Ecohydrology Research Group, Water Institute, Department of Earth and Environmental Sciences, University of Waterloo, 200 University Avenue, Waterloo, Ontario, Canada N2L 3G1. [2] Department of Geoscience, Environment & Society, Université Libre de Bruxelles, Brussels 1050, Belgium. [3] Department of Mathematics, College of Engineering, Mathematics and Physical Sciences, University of Exeter, Exeter EX4 4QE, UK. Correspondence and requests for materials should be addressed to T.M. (email: tmaavara@uwaterloo.ca).

Rivers act as reactive conduits connecting the continental and oceanic carbon (C) cycles: they annually deliver around 1.0 Pg C (83 Tmol) to the sea, about half under the form of total organic carbon (OC)[1–5]. Humans, however, have profoundly altered the balance between carbon fixation, mineralization and OC burial along the river continuum, not only by increasing the loadings of OC and nutrients to rivers but also through the massive building of dams[4,6–8]. Globally, there are over 16 million dams, with more than 50,000 large dams having heights exceeding 15 m (ref. 9). Over the next few decades, many new dams will be built, primarily for energy production. The number of large hydroelectric dams, which currently represent about 20% of large dams, is expected to double in the next 15–20 years[9,10]. All the available evidence suggests that river damming significantly changes riverine OC export to the ocean[1].

Upon dam closure, increased water residence time, improved light conditions, nutrient retention and sediment trapping in the impounded reservoir amplify primary productivity and promote the burial of autochthonous and allochthonous OC[11–14]. The construction of a dam also causes flooding and subsequent degradation of submerged biomass and soil organic matter within the reservoir[15–17]. These processes modify the downstream transfer of OC and nutrients, and thus the trophic state of the river system, as indicated by the decrease in $pCO_2$ observed in many rivers after dam closure[18–21]. However, the global-scale changes in riverine OC fluxes due to damming remain poorly quantified[11]. Only OC burial in the largest 600–700 reservoirs has been estimated[22], while global estimations of in-reservoir photosynthetic carbon fixation and mineralization are highly uncertain[1,4,16].

In this study, we present a worldwide analysis of decadal trends in riverine OC fluxes that explicitly accounts for reservoirs as a dynamic compartment of the C cycle on the continents[1,23]. We use a spatially explicit modelling approach to predict global in-reservoir primary production (P), mineralization (R) and burial of OC, from 1970 to 2050, to assess the evolving role of dams in the riverine export of OC to the oceans. We predict that dams decreased OC delivery to the oceans by 13% in 2000, with this value increasing to 19% by 2030. Our analysis further reveals that, during the period 1970–2030, worldwide reservoir respiration exceeds primary production (P<R), because of the continual addition of new dams. Should dam construction become negligible after 2030, dam reservoirs could globally become net autotrophic (that is, P>R) by the middle of the century.

## Results

**Dam databases**. The Global Reservoirs and Dams (GRanD) database provides the most comprehensive compilation of data on dams and reservoirs, including reservoir volume, surface area, water discharge through the dam and date of dam closure[9]. GRanD includes reservoirs with surface areas greater than $0.1 km^2$ and is near complete for reservoirs larger than $10 km^2$. The 6,862 reservoirs in GRanD[9] represent 77% of the estimated global reservoir volume of $8,069 km^3$. Small reservoirs with surface areas down to $0.0001 km^2$ make up the remaining volume. Our global estimates of carbon fixation, mineralization and burial for years 1970 and 2000 are based on the dams listed in GRanD (with closure dates preceding either 1970 or 2000), hence including a much larger number of reservoirs than previous estimates[24–26]. For the 2030 and 2050 projections, GRanD is augmented with the database compiled by Zarfl et al.[10] which contains over 3,700 hydroelectric dams with generating capacities ≥1 MW that are now under construction or planned to be completed by 2030. We assume that these dams will account for most of the increase in global reservoir volume in the first half of the twenty-first century.

**Modelling approach**. Dams under operation at the selected time points (1970, 2000, 2030 and 2050) are extracted from the dam databases. We assume that all the dams in the database of Zarfl et al.[10] will be completed by 2030, and that the ongoing dam building boom will end by 2030, beyond which no major further dam construction will take place. For each dam, we apply a mass balance model of in-reservoir OC cycling, following the approach previously applied to the nutrient elements silicon and phosphorus[27,28]. The allochthonous model explicitly distinguishes between particulate (POC) and dissolved OC (DOC) supplied to a reservoir, while the more labile autochthonous OC is represented as a single pool (Supplementary Fig. 1). Mineralization fluxes include the contributions of biological respiration processes, as well as photochemical degradation[6,29,30]. Annual primary productivity (P) in a reservoir is estimated as a function of nutrient availability, light intensity and mixing depth. Phosphorus is assumed to be the limiting nutrient for photosynthesis[31,32]: the bioavailable phosphorus in each reservoir is estimated using our previous model for phosphorus retention by dams[28]. For any given reservoir, the input fluxes of allochthonous POC and DOC supplied by the upstream catchment are obtained from the watershed yields estimated by the Global-NEWS model. The yields account for the effects of climate change (temperature and hydrology) as well as land use and land cover changes on POC and DOC loadings to rivers[22,33]. The OC mass balance model then computes annual removal fluxes of allochthonous and autochthonous OC by in-reservoir mineralization and burial, and via dam outflow (Supplementary Fig. 1). The parameter values for the various flux expressions are selected from probability distribution functions derived from literature data[28,34]. A Monte Carlo (MC) analysis of the model generates global relationships linking primary production, mineralization and burial of OC to reservoir hydrology (Supplementary Fig. 3). For the 2030 and 2050 projections, we implement the Global-NEWS OC yields for the Millennium Ecosystem Assessment (MA) scenarios[35]. Full details on the modelling approach, including parameterization, upscaling, model sensitivity and uncertainty assessment, are reported in the Methods section.

**Organic carbon cycling in reservoirs**. The results for 1970 and 2000 record the impacts of the first boom in dam construction that began after World War II[36]. During the last three decades of the twentieth century, the number of dams increased by more than 70%, while the total reservoir storage volume increased from about 3,500 to nearly $6,200 km^3$. The global input of allochthonous OC to reservoirs grew from $3.8 \pm 0.6$ Tmol per year (46 Tg C per year) in 1970 to $6.6 \pm 1.0$ Tmol per year (79 Tg C per year) in 2000 (Fig. 1), because of the doubling of the catchment area upstream of dams combined with the rising anthropogenic OC loading to rivers. At the same time, in-reservoir production of autochthonous OC grew from $0.65 \pm 0.20$ to $1.2 \pm 0.4$ Tmol per year (7.8–14.4 Tg C per year). As expected, the larger supply of allochthonous OC and higher in-reservoir C fixation caused a significant increase in the amount of OC trapped behind dams: from $1.1 \pm 0.3$ Tmol per year (13 Tg C per year) buried in 1970 to $2.2 \pm 0.6$ Tmol per year (26 Tg C per year) in 2000. In contrast, the estimated global OC mineralization rates in 1970 and 2000 are nearly identical, around 2.5–3.0 Tmol per year (30–36 Tg C per year). The reason is that in 1970 over 70% of in-reservoir mineralization was associated with the respiration of flooded biomass and soil OC in recently impounded reservoirs. In the 1960s alone, 1,405 GRanD dams were built worldwide[9]. However, because of the slow-down in dam construction toward the end of the twentieth century,

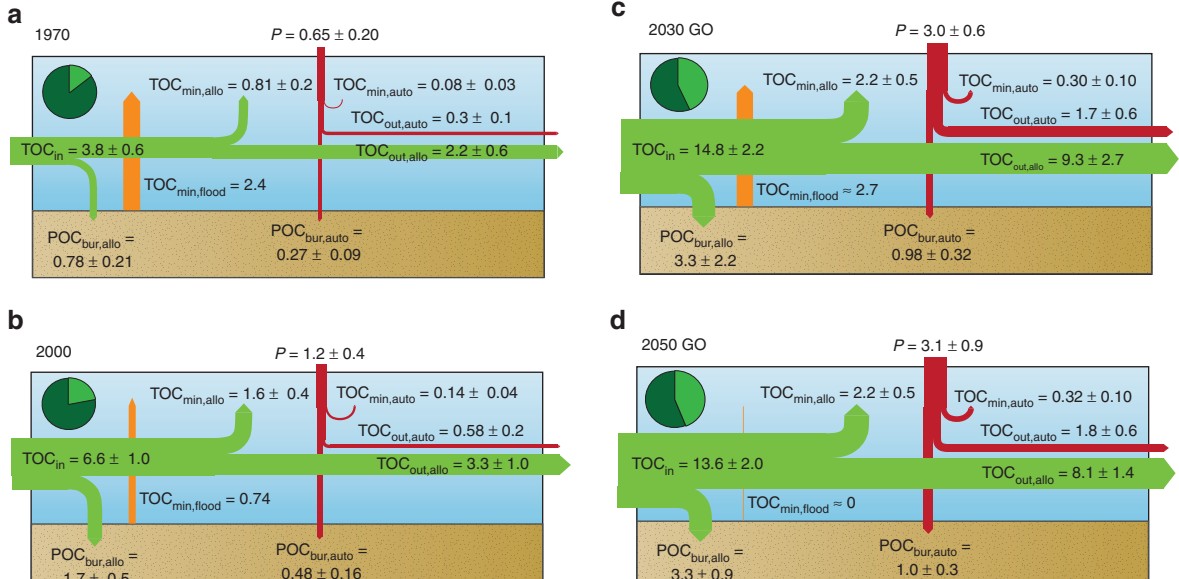

**Figure 1 | Global reservoir carbon budgets.** Globally integrated OC budgets of reservoirs in (**a**) 1970, (**b**) 2000, (**c**) 2030 and (**d**) 2050 (Global Orchestration, GO scenario). All fluxes are given in units of Tmol per year and rounded to two significant digits. Fluxes shown are the aggregated values for reservoirs worldwide. $TOC_{in}$: global influx of POC plus DOC to dam reservoirs (note: the routing procedure avoids double counting OC passing through cascades of dams, Supplementary Fig. 2). $P$: primary production. $TOC_{out}$: global efflux of POC plus DOC exiting dam reservoirs, without double counting TOC that passes through multiple dams. Subscripts: bur: burial; min: mineralization; flood: flooded terrestrial biomass and soil organic carbon; allo: allochthonous; auto: autochthonous; out: outflow. Errors assigned to $TOC_{in}$ reflect the uncertainties associated with the Global-NEWS model yield estimates. Other error estimates include those associated with the model parameterization, upscaling and errors in Global-NEWS (Methods section). Pie charts in top left of each panel represent the proportion of POC plus DOC loaded to global rivers that passes through at least one dam (shown in light green) before reaching the coastal zone.

with just 405 GRanD dams completed in the 1990s, flooded OC contributed only about 30% of reservoir mineralization in 2000.

The second boom in dam construction started shortly after the turn of the twenty-first century[10]. The new dams, however, tend to have shorter water residence times than those built in the last century because, for the majority of them, the primary purpose is hydropower production. Reservoirs of hydroelectric dams typically have smaller volumes and faster flows than the storage reservoirs of the twentieth century. In 2000, the median water residence time of reservoirs was about 8 months, for post-2000 dams it is on the order of 1 month. Of the dams under construction or planned to be completed by 2030, 7% have water residence times under 5 days, compared with just 3% in 2000. As a result, mean OC burial and mineralization efficiencies are predicted to be lower for the post-2000 dam reservoirs. For instance, we estimate that, in 2000, 40 and 12% of autochthonous OC produced in reservoirs were buried and mineralized, respectively, compared to 32 and 10% predicted for 2030. In turn, this implies an increase in the fraction of autochthonous OC transferred downstream of dams, from 48% in 2000 to 58% in 2030. The relatively small changes in OC fluxes between 2030 and 2050 are a direct consequence of the assumption that damming stops after 2030. Although it is unlikely that damming activity will cease abruptly, a slowing down in dam construction can be expected by 2030, at which time fragmentation by dams will already affect more than 90% of the total river volume on Earth[8].

**Production versus mineralization.** During the 1970–2030 period, reservoirs remain globally net heterotrophic with OC mineralization ($R$) exceeding primary production ($P$) (Fig. 1, Supplementary Table 6). The global $P/R$ ratio, however, is highly variable, with values ranging from 0.20 in 1970 to 0.58 in 2030.

The low $P/R$ value in 1970 reflects the large contribution to $R$ of mineralization of flooded terrestrial OC behind recently completed dams. Because flooded OC degrades on relatively short timescales ($\leq 10$ years), decreasing exponentially as the reservoir ages[16,17], global reservoir $P/R$ depends on the pace at which new dams are constructed. The 3,763 dams currently under construction or planned to be completed within the next two decades are expected to flood 80 Tmol (960 Tg C) of degradable OC (Methods section). The exact timing of the degradation of the flooded OC is difficult to predict as the completion dates for most of the new dams are uncertain[10]. We estimate that on average roughly 2.7 Tmol (32 Tg C) of flooded OC will be mineralized annually over the 2000–2030 period. If, as assumed in the 2050 projection, new dam construction becomes negligible after 2030, the mineralization of flooded OC will slow down rapidly and the global reservoir $P/R$ ratio could climb to values exceeding 1, reaching 1.24 in 2050. That is, reservoirs could become net autotrophic.

While mineralization of flooded OC has the greatest impact on the trophic state of a reservoir in the years following dam closure, mineralization of OC derived from the upstream catchment and that of OC produced *in situ* continue over the lifetime of the reservoir. On a global scale, mineralization of allochthonous OC exceeds that of autochthonous OC by a factor of 7–11, because the input of allochthonous OC is much greater than in-reservoir photosynthetic carbon fixation. Global reservoir OC mineralization, however, is primarily driven by the world's larger reservoirs that are usually associated with higher stream orders, that is, with the larger rivers that receive the flow from many tributaries (Supplementary Fig. 4). In fact, our model predicts that, on average, for Strahler stream orders smaller than five, in-reservoir productivity sustained by anthropogenic phosphorus loading exceeds mineralization, that is, $P/R > 1$.

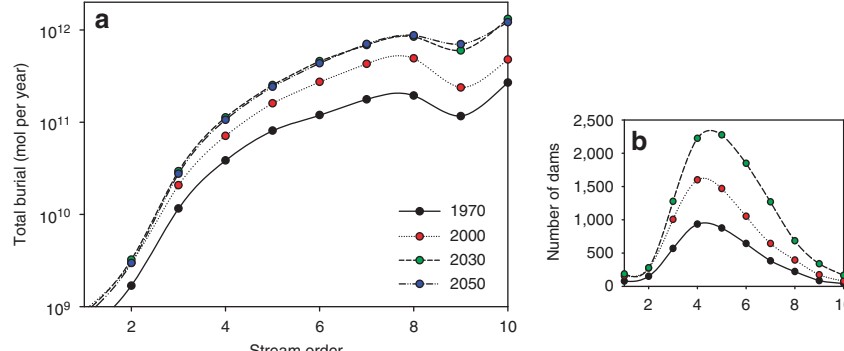

**Figure 2 | OC burial in reservoirs by stream order.** (**a**) Global OC burial (autochthonous + allochthonous) in dam reservoirs as a function of Strahler stream order. The 2030 and 2050 OC burial estimates correspond to the GO MA scenario. (**b**) Distribution of dams according to the Strahler stream order on which they are located. Note that the number of dams in 2050 is assumed to be equal to that in 2030. See Supplementary Table 3 for the assignment of Strahler stream orders.

The predicted net mineralization fluxes of reservoirs (that is, $R$–$P$) can be compared to reported carbon evasion rates, as the latter are primarily driven by in-reservoir OC mineralization. Barros et al.[16] estimate that reservoirs annually emit 4.2 Tmol of carbon to the atmosphere (51 Tg per year). Raymond et al.[37] propose a $CO_2$ evasion of 26 Tmol per year (320 Tg C per year) for lakes and reservoirs combined. Correcting the latter value using a regionalized lake to reservoir surface area ratio[26] then yields a global evasion flux from reservoirs alone of 3.3 Tmol per year (40 Tg C per year). These $CO_2$ evasion fluxes are of the same order of magnitude as our net mineralization fluxes for the period 1970–2000 (1.3–2.6 Tmol per year or 16–32 TgC per year). The $CO_2$ evasion flux from reservoirs of St. Louis et al.[37] is much larger: 23 Tmol per year (280 Tg C per year). Their estimate, however, is based on a limited set of $CO_2$ evasion rates that are scaled to a total reservoir surface area of 1.5 million km², that is, more than three times the surface area of the GRanD reservoirs.

**Organic carbon burial**. Around 75% of OC accumulating in reservoir sediments is allochthonous. Reservoirs along large, higher-order rivers are preferential sites of OC accumulation: we estimate that more than half of OC burial takes place in reservoirs on 8th order or higher rivers, although they only represent about 10% of all reservoirs worldwide (Fig. 2). These reservoirs tend to intercept the rivers carrying the highest OC loads. Globally, OC burial in reservoirs is expected to reach $4.3 \pm 1.2$ Tmol per year (52 Tg C per year) by 2030, that is, a fourfold increase relative to 1970 (Fig. 1). The increase in OC burial correlates with the growing number of dams and, to a lesser extent, the increasing OC loading to rivers. The latter primarily reflects changes in land use and land cover that cause increased soil erosion[1].

The predicted global net mineralization to burial ratio for reservoirs in 2000 is 1.1, which is comparable to the value of 1.5 of Cole et al.[4] for inland waters. Nonetheless, our global reservoir OC burial fluxes tend to fall below previously published values. For example, Mulholland and Elwood[12] estimate that reservoirs globally accumulate 17 Tmol per year of OC (200 Tg C per year) using burial rates from Europe and North America, Dean and Gorham[38] obtain a value of 13 Tmol per year (160 Tg C per year) by extrapolating average values of sedimentation rate, OC concentration and bulk density to the global reservoir surface area, while Stallard's[39] similarly large estimate of 24 Tmol per year (290 Tg C per year) is calibrated using parameters from reservoirs in the United States. The discrepancy is likely due to two reasons. First, our fluxes only

account for reservoirs in the GRanD[9] and Zarfl et al.[10] databases and, hence, ignore the contributions of the large number of small reservoirs. Second, the earlier estimates rely on empirical extrapolations from data on small numbers of reservoirs with uneven geographical coverage, mostly concentrated in temperate climate zones[12,40]. As a result they probably over-predict burial OC rates, because they do not account for the high mineralization rates in the tropics. Further note that the C burial rate of 1–3 Pg C per year (83–250 Tmol per year) for reservoirs reported by Syvitski et al.[5] includes inorganic carbon, while our estimates only account for OC burial.

**Regional hotspots**. From 1970 to 2000, reservoirs in the Mississippi, Niger and Ganges River basins buried most OC (Fig. 3): together, they accounted for 31 and 25% of global OC burial by dams in 1970 and 2000, respectively (Supplementary Data 1). In 1970, the highest in-reservoir OC elimination occurred in the Mississippi River basin, where 192 Gmol of OC (2.3 Tg C per year) were buried and 305 Gmol (3.7 Tg C per year) mineralized annually. By 2000, OC mineralization in reservoirs of the Paraná and Zambezi River basins overtook the Mississippi River basin. The estimated 632 Gmol per year (7.6 Tg C per year) mineralized in 70 dam reservoirs along the Paraná River and its tributaries in the year 2000 reflect degradation of flooded material following construction of new dams at the end of the twentieth century, including the 11 km long Eng Sérgio Motta Dam. The high mineralization in the Zambezi basin is due in large part to Lake Kariba, the world's largest reservoir by volume. Other hotspots of reservoir OC burial include the basins of the Danube in Europe, the Ganges and Mekong in central and Southeast Asia, the Yenisei in Russia, the large Chinese rivers and the Tocantins in South America. The OC burial hotspots generally coincide with OC mineralization hotspots (Fig. 3). One notable exception is the Mackenzie River basin where in 1970 relatively low-OC burial fluxes coexisted with high-OC mineralization rates. The latter are explained by the large pulse in mineralization of flooded soil OC and biomass in Williston Lake, the seventh largest reservoir globally by volume, following completion of the W.A.C. Bennett Dam in 1968.

Damming is increasingly focused in river catchments of Asia, South America, Africa and the Balkans (Supplementary Fig. 5). The Amazon, Yangtze and Ganges basins are expected to remain the primary hotspots for reservoir OC burial and mineralization. In the Amazon basin, with the planned completion of 184 new dams by 2030, OC burial will increase 38-fold to 292 Gmol per

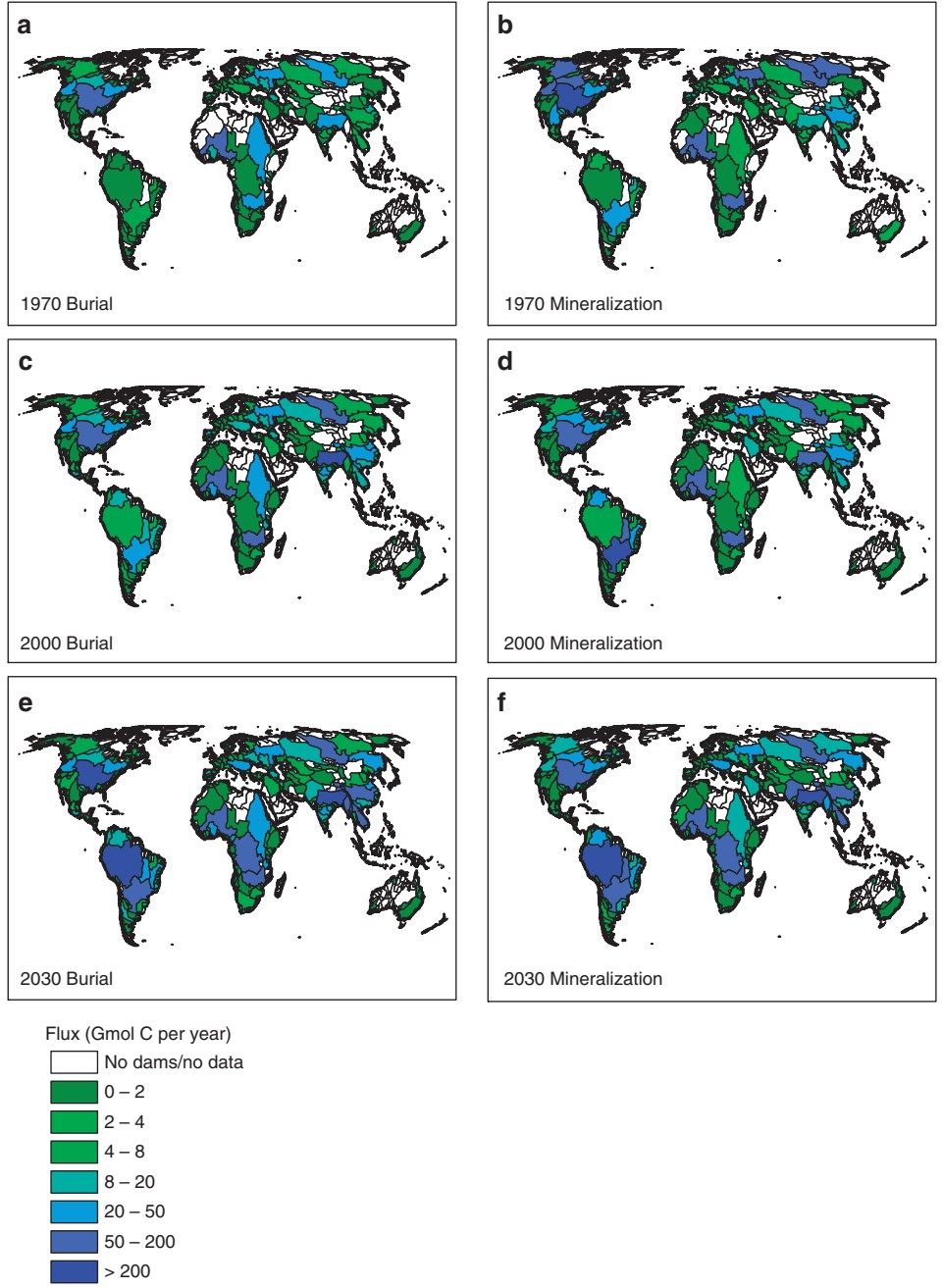

**Figure 3 | Reservoir mineralization and burial in the world's river basins.** Mineralization and burial fluxes of OC in reservoirs of the main river basins of the world, for 1970 (**a**,**b**), 2000 (**c**,**d**) and 2030 GO scenario (**e**,**f**), in units of Gmol per year. To estimate the 2030 mineralization fluxes of flooded OC for dams without defined completion dates or completion dates that can be estimated based on start dates ($n = 2,925$), we randomly assigned completion dates to evenly space dam closures over time between 2000 and 2030. The uncertainty associated with this procedure does not substantially affect the 2030 spatial trends.

year (3.5 Tg C per year), or 7% of global reservoir OC accumulation. For the Yangtze River basin, the estimated 2030 reservoir OC burial and mineralization fluxes are 172 and 152 Gmol per year (2.1 and 1.8 Tg C per year). The corresponding total OC removal by reservoirs ($172 + 152 = 324$ Gmol per year) agrees well with the reduction in riverine OC flux since the 1950s of $408 \pm 158$ Gmol per year, derived independently by Li *et al.*[42] using sediment core data from the lower reaches of the Yangtze River. These authors ascribe the drop in OC flux to the construction of dams in the Yangtze River basin, which started in the 1950s. Note that we use our estimate for 2030 rather

than 2000, to account for the very large dams built after 2000, including the Three Gorges Reservoir dam.

The Congo (Zaire) basin may experience a major surge in dam construction over the next 20–30 years. Of the 15 new dams planned or under construction, the most notable is the proposed Grand Inga Dam, which could surpass the Three Gorges Dam in terms of hydroelectric generating capacity. When built, the new reservoirs in the Congo basin would collectively bury and mineralize on the order of 68 and 113 Gmol per year (0.82 and 1.4 Tg C per year), respectively. New dams in the basins of the Mekong, Paraná, Salween and Tocantins rivers are also predicted

to contribute substantially to reservoir OC respiration in 2030, largely as a result of the degradation of recently flooded soil and biomass OC (Fig. 3, Supplementary Data 1). Increases to OC mineralization fluxes in tropical reservoirs are also expected between 2000 and 2030 due to warmer water temperatures, particularly in the 0–10°S latitude band (Supplementary Fig. 6).

Most existing regional carbon flux estimations for inland water bodies do not separate dam reservoirs from lakes. Hence, the corresponding fluxes provide upper limits to which we can compare our values for reservoirs alone. For instance, the U.S. Climate Change Science Program[40] estimate that inland waters in North America buried 1.9 Tmol per year (23 Tg C per year) in 2003. As expected, our reservoir-only OC burial flux for North America in 2000 is smaller, on the order of 0.3 Tmol per year. Similarly, in Russia, where reservoirs make up around 12% of the total inland water surface area[20,26], our estimate of in-reservoir net mineralization represents about 13% of the value of 0.98 Tmol per year (12 Tg C per year) reported by Dolman et al.[41] for all Russian inland waters together.

**Organic carbon export to the global coastal zone**. Dam reservoirs modify the riverine export of OC to the coastal ocean: primary production adds new OC to the river system, while burial and mineralization eliminate OC. At the global scale, dams represent a net sink for riverine OC, as burial and mineralization of allochthonous and autochthonous OC together exceed in-reservoir production for the entire 1970–2050 period. The evolution over time of the global reservoir OC sink is closely related to the changing fraction of the world's river catchment area located upstream of dams: 18% in 1970, 27% in 2000, and 36% in 2030 and 2050 (Supplementary Table 6), which in turn determines the proportion of the global OC load that enters dam reservoirs (Fig. 1). As more and more catchment area leads into dam reservoirs, an increasing proportion of riverine OC is eliminated. In particular, the larger dams in lowland areas near the coast have the greatest potential to eliminate OC as they intercept the loads generated throughout the catchment. These dams thus play a disproportionate role in OC burial and mineralization along the aquatic continuum.

According to our calculations, in 2000, dams lowered the OC export to the global coastal zone via rivers by 13%: 7% through sequestration of OC in reservoir sediments and 6% through in-reservoir mineralization (Supplementary Table 6). For comparison, for the same period, dams decreased riverine export of reactive silicon and phosphorus by only 5 and 12%, respectively[27,28]. All other factors equal, by eliminating OC more efficiently than nutrients, river damming should enhance carbon fixation over mineralization in the receiving waters. With the ongoing rise in the number of dams, riverine OC export in 2030 could be 19% lower than if no new dams were built after 2000 (12% due to burial and 7% to mineralization). Therefore, in all but one of the MA scenarios, the current boom in dam construction could even cause a net decrease in OC export to the coastal zone between 2000 and 2030, despite the increased anthropogenic loading of OC to watersheds (Supplementary Table 6). Through its large impact on the delivery of riverine OC, damming therefore represents a major anthropogenic forcing on the trophic conditions and carbon balance of the coastal ocean.

## Methods

**Modelling approach**. In-reservoir OC transformations and burial rates are simulated using a simple, biogeochemical mass balance (or box) model (Supplementary Fig. 1). The parameters of the kinetic expressions representing the in-reservoir processes are assigned probability density functions (PDF) that take into account each parameter's inter- and intra-reservoir variability at the annual timescale. The PDFs are derived from available data on photosynthetic C fixation,

OC mineralization and OC burial in lentic systems. A similar approach was previously applied to predict global-scale phosphorus and nutrient silicon retention in dam reservoirs[27,28].

A virtual database of model reservoirs is generated by performing 6,000 MC simulations with rate parameters selected randomly from their corresponding PDFs. From this virtual database, global relationships are derived that predict burial, mineralization and photosynthesis rates from the reservoir's hydraulic residence time. The global relationships are applied to a real-world database that includes existing reservoirs (GRanD)[9] in 1970 and 2000, and reservoirs under construction or planned to be completed by 2030 (ref. 10). Together with the DOC and POC loads to reservoirs obtained from the Global-NEWS model, the burial, mineralization and photosynthesis fluxes in the reservoirs of all major watersheds worldwide are then simulated for four selected time points (1970, 2000, 2030 and 2050). The MC analysis also yields estimates of the uncertainties on the OC fluxes associated with parameter variability.

The Global-NEWS model is well suited for the proposed modelling approach: it differentiates between POC and DOC, it implicitly accounts for in-stream OC losses, and it has been used to hindcast nutrient loads to watersheds in the years 1970 and 2000, and to forecast the 2030 and 2050 loads according to the MA scenarios[22,35]. Global-NEWS predicts sediment, OC and nutrient yields based on land use and land cover (for example, wetlands, cropland and grasslands), climate variables (including temperature and precipitation), geomorphological parameters (including slope and lithology) and anthropogenic alterations (including consumptive water usage). The MA scenarios are storylines for a future world that will become either more globalized (Global Orchestration (GO) and TechnoGarden, TG) or regionalized (Adapting Mosaic, AM, and Order from Strength, OS), and take either a proactive (TG and AM) or reactive approach to environmental management (GO and order from strength, OS)[35]. Each of the MA scenarios assumes changing land uses, climate regimes and anthropogenic perturbations, which in turn modify the fluxes of sediments, OC and nutrients delivered to river systems.

A caveat of Global-NEWS is the lack of representation of extreme hydrological events with a low recurring probability but potentially large contributions to long-term riverine fluxes. These events are not captured by the Global-NEWS model itself or by the observed data it was trained on. These events can involve geomorphologic processes such as landslides, which mobilize the carbon stored in the entire soil horizon plus the standing biomass. Such events, which can be related to extreme weather phenomena (for example, tropical cyclones) or earthquakes, have been described for steep catchments in the sub-tropics (for example, Taiwan[43]) and temperate regions (for example, New Zealand[44]). Their contributions to OC mobilization at the global scale remain to be quantified.

In our modelling approach, reservoir infilling due to sedimentation is not taken into account. Vörösmarty et al.[45] estimate that sedimentation has reduced the volume of reservoirs in the United States by up to 2 km³, that is, only ∼0.2% of the total US reservoir volume. While reservoir infilling may vary significantly from one reservoir to another, the effect of sediment accumulation on water residence time likely represents a relatively minor source of uncertainty on the impact of dams on the global riverine OC flux.

**Mass balance model**. The mass balance model for in-reservoir allochthonous and autochthonous OC cycling is shown in Supplementary Fig. 1. Riverine inflow fluxes of allochthonous POC and DOC are calculated by multiplying inflow concentrations (mol km⁻³) by the river discharge ($Q$, km³ per year) through the dam. Assuming inflow of water to a reservoir equals the outflow through the dam, values for $Q$ are retrieved from the GRanD database[9], augmented with the database of hydroelectric dams of Zarfl et al.[10] for the 2030 and 2050 projections. Outflow fluxes of OC through the dam are obtained by multiplying the in-reservoir POC and DOC masses (mol) by the flushing rate, $\rho$ (per year), which is equal to $Q/V$ where $V$ is the reservoir volume in km³. The key role of $\rho$ (or its inverse, the water residence time) in material mass balances of reservoirs is well-established[6,24,27,46].

Except for primary production, the fluxes ($F$, mol per year) describing the in-reservoir OC processes assume first-order kinetics with respect to the mass ($M$, mol) of the OC pool from which the flux originates:

$$F = k \times M \qquad (1)$$

where $k$ is the first-order rate constant (per year). Primary production of autochthonous POC ($P$, mol per year) is assumed to be phosphorus limited[31,32,47], according to

$$P = P_{max} \frac{[\text{TDP}]}{K_s + [\text{TDP}]} \qquad (2)$$

where $P_{max}$ is the maximum value of $P$ under nutrient saturated conditions, [TDP] the concentration of total dissolved phosphorus in the reservoir and $K_s$ the half-saturation TDP concentration. For each reservoir and time point, [TDP] is extracted from the previously published global reservoir phosphorus model[28]. The mass balance equations for the various OC pools are solved using the Runge–Kutta 4 integration scheme with 0.01-year time steps, and run for the length of time since dam closure (that is, if the dam is 10 years old, the model is run for 1,000 time steps).

**Model parameterization.** For each parameter value a PDF is derived from data reported in the literature (Supplementary Table 1). For the river inflow DOC concentration we impose the gamma distribution proposed by Sobek et al.[48] based on data for 7,500 lakes. This is justified as the mean and range of DOC concentrations in rivers are comparable to those observed in lakes[49,50]. For the inflow POC concentrations, a generalized Pareto distribution yields the highest log-likelihood when fitted to the POC concentrations derived by applying the Global-NEWS loads and discharge values to the GRanD reservoirs[9,22]. Similarly, the PDFs for reservoir volume, discharge, latitude, altitude and age (that is, years since dam closure) are the functions that were found to best fit the observed distributions in the GRanD database (Supplementary Table 1). Reservoirs in GRanD further yield the following relationship between surface area (SA, $km^2$) and volume:

$$SA = 42.264 \times V^{0.8183} \qquad (3)$$

In the MC analysis, SA values from equation (3) are multiplied by a random number between 0.1 and 10, thus allowing deviations in SA by ± one order of magnitude from equation (3). We compute SA from $V$, rather than vice versa, because $V$ is the primary size variable used to calculate the water residence times in the up-scaling procedure.

The temperature dependence of the allochthonous POC and DOC mineralization (biological respiration and photochemical degradation) rate constants, $k_{min,allo}$, is assumed to follow the same expression as that derived by Hanson et al.[29,30] from a compilation of ecosystem-scale mineralization rates of allochthonous DOC in lakes:

$$k_{min,allo} = k_{20} \times \theta^{(T-20)} \qquad (4)$$

where $T$ is the average water temperature (°C), $k_{20}$ is the rate constant at 20 °C, and $\theta$ is a dimensionless coefficient equal to 1.07 (ref. 29). The PDF of $k_{20}$ in Supplementary Table 1 is directly derived from the data of Hanson et al.[29,30]. While the same $k_{20}$ PDF is applied to both allochthonous DOC and POC[29], the $k_{20}$ values for POC and DOC are allowed to vary independently in the MC analysis. This approach allows for reactivities of allochthonous POC that in some cases are higher, in others lower, than those of allochthonous DOC, as has been reported for lentic systems[51,52].

In-reservoir produced autochthonous OC tends to be more labile than allochthonous OC[53,54]. The $k_{20}$ value for autochthonous OC is therefore multiplied by a variable scaling factor. An average three-times higher autochthonous $k_{20}$ yields the best fit of our model-predicted total carbon mineralization rate constants to the rate constants compiled by Catalan et al.[17] across a wide range of aquatic ecosystems and climate zones. To capture the range of the Catalan et al.[17] rate constant data (with the exception of four outliers, three of which are from the same small streams), the autochthonous scaling factor is assumed to follow a normal distribution between 1 and 6 (Supplementary Table 1). With this range of the scaling factor, the model-generated total OC mineralization rate constants reproduce the range of observed rate constants of Catalan et al.[17] (Supplementary Fig. 7). Because of its greater lability, autochthonous OC exported through a dam is assumed to be mineralized before reaching a downstream reservoir.

The reservoir water temperature $T$ is computed as a function of latitude and altitude following Hart and Rayner[55]. For 2030 and 2050, the average air temperature increases predicted by Fekete et al.[33] for each individual MA scenario are converted to reservoir water temperature increases using the relationship of Lauerwald et al.[20], and added to the temperature predicted with the Hart and Rayner equation (Supplementary Table 6).

The half-saturation constant $K_s$ in equation (2) is represented by a uniform PDF ranging from $2.0 \times 10^7$ to $7.0 \times 10^8$ mol $km^{-3}$ in units of total dissolved phosphorus[56–64]. The maximum primary production $P_{max}$ in equation (2) is estimated with:

$$P_{max} = B \times P_{Chl} \times V \times M \times D_f, \qquad (5)$$

where $B$ is the average annual depth-integrated chlorophyll concentration in the reservoir (mg Chl-a $km^{-3}$), $P_{Chl}$ is the maximum, annually integrated, chlorophyll-specific carbon fixation rate (mol C (mg Chl-a)$^{-1}$ per year), $M$ is a dimensionless metabolic correction factor for water temperature and $D_f$ is the yearly proportion of ice-free days. The value of $P_{Chl}$ is fixed at 0.15 mol C (mg Chl-a)$^{-1}$ per year, which is equivalent to 2.5 g C (g Chl-a)$^{-1}$ h$^{-1}$, assuming an annual average of 12 h daylight per 24 h (refs 65–68); $M$ is equal to 1 at water temperatures $\geq 28$ °C, and decreases at lower temperatures with a $Q_{10}$ of 2 (ref. 69); the duration of ice cover required to estimate $D_f$ is calculated from latitude following Williams et al.[70]; $B$ is updated in each model iteration using the relationship provided by Reynolds[71]:

$$B = \left(\frac{1}{k_c}\right) \left[ 0.75 \left(\frac{PP}{RP}\right) \left(\frac{DL}{24}\right) \ln \left(\frac{0.7 I_{o,max}}{0.5 I_k}\right) \left(\frac{1}{Z_{mix}}\right) - \left(K_{dw} + K_{dp} + K_{dg}\right) \right] \qquad (6)$$

where $k_c$ is the absorbance of photosynthetically active radiation (PAR) per unit of chlorophyll (fixed at 0.014 $m^2$ (mg Chl-a)$^{-1}$ globally[69]), PP/RP is the ratio of maximum gross photosynthesis to algal respiration per unit chlorophyll, fixed at 15 (ref. 72), DL is the hours of daylight, fixed annually at 12 h per day[71], $I_{o,max}$ is the maximum site-specific PAR ($\mu$mol m$^{-2}$ s$^{-1}$), $I_k$ is the PAR at the onset of photosaturation, fixed at 120 $\mu$mol m$^{-2}$ s$^{-1}$[71], $Z_{mix}$ is the reservoir mixing

depth (m) and $K_{dw} + K_{dp} + K_{dg}$ is the nonalgal PAR attenuation (m$^{-1}$). The value of $I_{o,max}$ is calculated for each reservoir based on the annually averaged latitude-specific values provided by Lewis[69] assuming linear interpolation between the latitudinal bands provided; $Z_{mix}$ is estimated based on the empirical relationship with lake fetch[69], where fetch is assumed equal to the diameter of a circle with the same area as the reservoir; PAR attenuation in pure water, $K_{dw}$, is fixed at 0.13 m$^{-1}$, PAR attenuation by inorganic suspended particulate matter (tripton), $K_{dp}$, is fixed at 0.06 m$^{-1}$, and that by dissolved organic matter (gilvin), $K_{dg}$, is calculated for each reservoir with the relationship proposed by Lewis[69]:

$$\ln K_{dg} = -4.44 + 1.80 \times \ln[DOC] - 0.149 \times (\ln[DOC])^2 \qquad (7)$$

where [DOC] in p.p.m. is generated using the same PDF as for inflow [DOC][48]. A $t$-test shows that $B$-values generated in the MC analysis are statistically indistinguishable ($P < 0.05$) from values found in the literature[65,69,71].

The burial rate constant, $k_{bur}$, is an effective parameter describing the long-term retention of POC with the sediments accumulating in the reservoir, that is, the POC that is not remineralized or otherwise remobilized and exported over the reservoir's lifetime. The value of $k_{bur}$ aggregates all the factors controlling the POC burial efficiency other than the water residence time, including sedimentation rate, reservoir morphology (which controls deposition patterns), oxygen exposure time, temperature and sediment resuspension events[53]. The upper and lower bounds for the uniform $k_{bur}$ distribution, 1 and 15 per year respectively, are based on published rate constants[73] and values back-calculated from global and regional data sets of OC accumulation rates in lakes and reservoirs[38,53,74]. More studies quantifying burial rate constants in a diversity of reservoir settings will be needed to further delineate the form of the $k_{bur}$ PDF. The rate constant for allochthonous POC solubilization to DOC is kept fixed at 0.1 per year, as the model is insensitive to this parameter (see Model sensitivity and uncertainty section)[75].

**Global upscaling.** Global regression relationships are obtained from the MC analysis by carrying out 6,000 iterations with the 1970 and 2000 parameter values each, plus 6,000 additional iterations for each of the MA scenarios in 2030 and 2050 taking into account the projected changes in air temperature. For all time points and MA scenarios considered, the water residence time, $\tau_r$, provides the best predictor for the fate of allochthonous OC entering a reservoir. The in-reservoir burial and mineralization fluxes, $F_{i,bur}$ and $F_{i,min}$, produced by the MC simulations are fitted to

$$F_{i,bur} = F_{i,in} \times a_i \left[ 1 - \frac{1}{1 + \alpha_i \times \tau_r} \right] \qquad (8)$$

and

$$F_{i,min} = F_{i,in} \times b_i \left[ 1 - \frac{1}{1 + \beta_i \times T \times \tau_r} \right] \qquad (9)$$

where the subscript $i$ stands for allochthonous POC or DOC (POC only in the case of burial), in stands for inflow, $T$ is average annual reservoir water temperature (°C), $\alpha_i$ and $\beta_i$ are first-order rate coefficients describing loss due to burial and mineralization, respectively, and $a_i$ and $b_i$ are dimensionless coefficients. The best-fit values of $\alpha_i$, $\beta_i$, $a_i$ and $b_i$ are given in Supplementary Table 2. Note that equations (8) and (9) are formally similar to the equation derived by Vollenweider[46] to describe phosphorus cycling in lakes, though scaled with $a_i$ and $b_i$ to account for the separation of loss fluxes into mineralization and burial, rather than lumped together as in Vollenweider's derivation.

The inflows of allochthonous DOC and POC, $F_{i,in}$, are estimated by spatially overlaying the GRanD reservoirs[9] onto the Global-NEWS watersheds. Global-NEWS is used because it generates spatially explicit riverine OC and phosphorus yields for the four MA scenarios. The projected MA global average air temperature increases are 0.91–1.09 °C in 2030 and 1.29–2.11 °C in 2050, relative to 2000. These projections are similar to the temperature increases of the Representative Concentration Pathways scenarios RCP4.0 and RCP6.5 (ref. 76). Reservoir water temperature, $T$, is calculated for each reservoir in GRanD using the relationship with latitude and altitude derived by Hart and Rayner[55]. For the 2030 and 2050 scenarios, an increase in water temperature is added to each reservoir based on the average latitude-specific temperature increases predicted by Fekete et al.[33].

All reservoirs are spatially routed to account for dams that occur longitudinally in series, and dams on tributaries as illustrated in Supplementary Fig. 2. The total influx (mol per year) of allochthonous POC or DOC to a reservoir $k$ is then given by:

$$F_{k,in} = \sum_1^n F_{k-1,out} + \left( W_k - \sum_1^n W_{k-1} \right) \times Y_k \qquad (10)$$

where $\sum_1^n F_{k-1,out}$ is the sum of the fluxes of allochthonous POC or DOC leaving all dams immediately upstream of reservoir $k$ on tributaries 1 to $n$, $W_k$ is the total upstream watershed area ($km^2$), $\sum_1^n W_{k-1}$ is the sum of the upstream watershed areas of dams $k$-1, and $Y_k$ is the POC or DOC yield (mol $km^{-2}$ per year) in the undammed, $\left( W_k - \sum_1^n W_{k-1} \right)$, watershed area downstream of dams $k$-1. In other words, the OC loads leaving all upstream reservoirs are added to the OC load entering the river from its undammed upstream watershed area.

In Global-NEWS, carbon and nutrient loads from landscapes to rivers are empirically adjusted to match the fluvial DOC and POC export fluxes measured at

the mouths of large rivers. To reconcile our estimates of OC elimination in reservoirs with the observed OC export fluxes to the coastal ocean, the Global-NEWS allochthonous OC loads to rivers are recalibrated. As initial hypothesis we impose the original Global-NEWS DOC and POC loads to calculate in-reservoir elimination fluxes and export fluxes to the coastal ocean. The loads are then iteratively adjusted until they reproduce the Global-NEWS coastal OC export fluxes for 1970 and 2000. For the 2030 and 2050 scenarios, which are based on projections rather than calibrated to data, we apply the same correction factor to any given watershed as that for 2000. On average, the revised OC loads to watersheds differ only by ± 1.3% from the original Global-NEWS estimates.

The combined outputs of the MC analyses of reservoir OC and phosphorus models yield the following relationship between in-reservoir primary production, $P$, and the inflow flux of total dissolved phosphorus, $TDP_{in}$ (mol per year):

$$P = e^{10.5042 \pm 0.0939}\, TDP_{in}^{0.5938 \pm 0.0085} \quad (R^2 = 0.48) \qquad (11)$$

where the uncertainty associated with each parameter corresponds to ± 1 s.d. (see Model sensitivity and uncertainty section). Equation (11) is then used in relationships of the form of equations (8) and (9) to describe the fractions of in-reservoir produced autochthonous OC that are buried and mineralized:

$$F_{j,bur} = P \times a_j \left[ 1 - \frac{1}{1 + \alpha_j \times \tau_r} \right] \qquad (12)$$

and

$$F_{j,min} = P \times b_j \left[ 1 - \frac{1}{1 + \beta_j \times T \times \tau_r} \right] \qquad (13)$$

where the subscript $j$ stands for burial and mineralization of autochthonous OC. The values of the parameters $\alpha_j$, $\beta_j$, $a_j$ and $b_j$ are listed in Supplementary Table 2.

The initial amount of degradable soil and biomass OC flooded after closure of a dam, $TOC_0$, is estimated from global soil[77] and biomass carbon[78] maps, scaled to the surface area of the reservoir. The decay of the flooded OC follows:

$$TOC(t) = TOC_0\, e^{-k_{min,flood} t} \qquad (14)$$

where $t$ is the number of years since dam closure. We assume the same temperature function (that is, $\theta^{(T-20)}$) for $k_{min,flood}$ as in equation (4) and adjust the pre-function coefficient $k_{20}$ so as to reproduce the decay timescale of flooded soil organic matter typically observed after dam closure (10–15 years)[16,79]. The resulting expression is then:

$$k_{min,flood} = 2.21 \times 1.07^{(T-20)} \qquad (15)$$

The Strahler stream orders of the GRanD reservoirs (Supplementary Table 3) are estimated with the empirical scaling law relating stream order to catchment area derived by Lauerwald et al.[20]. This scaling law is derived from the HydroSHEDS stream network for third order stream and higher[80], and extrapolated for the two lowest stream orders that are not represented in HydroSHEDS[20]. To test the validity of the scaling law, we have compared calculated stream orders with actual stream orders from the HydroSHEDS stream network for all dammed streams and rivers in Europe ($n = 2,192$) and South America ($n = 1,602$), yielding statistically significant $R^2$ values of 0.90 and 0.87, respectively.

**Model sensitivity and uncertainty.** Sensitivity analyses are performed separately for in-reservoir allochthonous and autochthonous OC cycling. The effects on burial and mineralization fluxes of changing one parameter value at the time are summarized in Supplementary Tables 4 and 5. In most cases, the parameter is varied ± 10% from the assigned 'default' value in 100 iterations, the results of which are compared to the burial and mineralization fluxes obtained using the default value. Reservoir volume and discharge (and hence water residence time) are varied according to the distributions in Supplementary Table 1. Sensitivity to initial conditions is assessed by comparing the results of model runs with the initial reservoir DOC, POC and TOC masses set equal to either 0 or $1 \times 10^6$ mol. Two reservoir ages are tested: 10 and 40 years. Sensitivity to a 1 °C increase in global air temperature is determined, which is equivalent to a water temperature increase of 0.82 °C. Not surprisingly, allochthonous and autochthonous OC burial fluxes are most sensitive to the rate constant of burial $k_{bur}$ while mineralization fluxes are most sensitive to $k_{20}$ and latitude, given the dependence of $k_{min}$ on temperature, which is in turn related to latitude, and to a lesser degree, altitude (Supplementary Tables 4 and 5).

A bootstrap analysis is used to estimate the uncertainties associated with primary productivity $P$ calculated with equation (11): sampling with replacement was conducted drawing 5,000 samples from the 6,000 model runs generated in the MC analysis, and fitted to a power law as in equation (11). After 5,000 iterations, the bootstrap analysis yields the s.d. estimates for each parameter in equation (11). When scaled up globally, the uncertainty on $P$ translates into uncertainties in the mineralization and burial fluxes of autochthonous OC of ± 15%.

The total uncertainties in the global reservoir OC burial and mineralization are assessed as follows. Burial and mineralization fluxes predicted by the 6,000 MC iterations are binned according to water residence times as shown in Supplementary Fig. 3. Gamma functions are fitted to the distributions of fluxes in each bin. A second MC analysis is then carried out in which, for each GRanD

reservoir, the burial and mineralization fluxes are randomly selected from the gamma functions. A total of 20 simulations yields ± 8% s.d. on the global OC fluxes for allochthonous DOC mineralization, ± 15% for allochthonous POC mineralization and ± 12% for allochthonous burial. For autochthonous OC fluxes, this analysis yields ± 2% s.d. for mineralization and ± 3% s.d. for burial fluxes, respectively. Combined with the ± 15% error associated with the Global-NEWS OC loads to rivers, we estimate uncertainties on the order of ± 23%, ± 30% and ± 27%, for allochthonous DOC mineralization, and allochthonous POC mineralization and burial, respectively. For the global values of autochthonous OC burial and mineralization, we estimate an uncertainty on the global fluxes of ± 32% for mineralization and ± 33% for burial. The higher uncertainties for autochthonous OC reflect the uncertainty in estimates of $P$ (see above). Note that the estimated uncertainties in the burial and mineralization fluxes are those associated with the mass balance modelling approach, and do not account for inaccuracies and omissions in the GRanD or future dam databases.

**Data availability.** Reservoir burial and mineralization fluxes of OC for individual river basins, in 1970, 2000 and 2030 (GO scenario), are given in Supplementary Data 1. For access to computer code, please contact Taylor Maavara.

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

## Acknowledgements

Funding was provided by the Canada Excellence Research Chair (CERC) program, a Natural Sciences and Engineering Research Council of Canada (NSERC) postgraduate scholarship to T.M., and the European Union's Horizon 2020 research and innovation program under Marie Sklodowska-Curie grant agreement no. 643052 (C-CASCADES project). R.L. acknowledges support from the Université Libre de Bruxelles, through the Bureau des relations internationales (BRIC), the French National Research Agency ('Investissement d'Avenir', ANR-10-LABX-0018), and the European Union's Horizon 2020 research and innovation program under grant agreement no.703813 for the Marie Sklodowska-Curie European Individual Fellowship 'C-Leak'. We are particularly grateful to Christiane Zarfl for sharing her database of future dams.

## Author contributions

T.M. and P.V.C. designed the research questions and overall modelling approach. T.M. and R.L. developed and coded the model, with input from P.V.C. and P.R. All authors contributed to writing the paper.

## Additional information

**Competing interests:** The authors declare no competing financial interests.

