## [Peer Review File · Nature Communications]

Reviewers' comments:

Reviewer #1 (Remarks to the Author):

- A. The summary of these results cannot at this time be evaluated without further justification of the assumptions made in the model and if these other issue raised below can be addressed.
- B. This is highly original and a very worthwhile exercise, we do need to try an make such estimates, but their assumptions are not realistic.
- C. The overall approach seems fine, the authors' have a good reputation in modeling.
- D. Not good, the assumptions and lack of consideration of other important processes in the watershed have likely led to serious flaws in the estimates.
- E. same as above.
- F. see below
- G.yes fine
- H. Well-written but serious issues in what all this means with the problematic assumptions.

This manuscript entitled "Global perturbation of organic carbon cycling by river damming" provides a well-structured modelling approach to quantify the impacts of river damming on the global carbon cycle. Although this model is sound in theory, the k-values for the fluxes need to be altered or justified, this is a major flaw as many are not well supported as I will describe below. I understand that the model needs to be simplified, but these are key processes that greatly impact the model and what is learned about this perturbation to the carbon cycle.

General Comments:

1. The assumption that POC and DOC have the same reactivity and the assumption that autochthonous OC is 3X more reactive than allochthonous OC need to be further supported with data from the water column. The Gudasz et al (2010) data, used to derive equation 4, includes data from lakes with varying ratios of allochthonous to autochthonous inputs. Therefore, the reactivity of autochthonous and allochthonous OC should be the same in this model if using the Gudasz data.
2. Can he authors provide more evidence to support excluding photochemistry in the model?
3. It would be useful if the fluxes were reported in terragrams instead of terramoles for comparison with other studies.
4. Can the authors account for riverine loss of OC during transport between reservoirs?
5. Can the authors test the model on a specific reservoir or river that is well studied to further ground-truth the model?

All of these issues. Except number 3 are serious problems that need justification and elaboration before the model can be fully evaluated.

Specific Comments:

Line 111, 115: The fluxes presented in text for 1970 and 2000 do not match the fluxes in Figure 1.

Methods Line 227: Could the author provide evidence/references for the use of equation 4 for calculating the k value for TOC decay in flooded soils?

In conclusion, I think that the importance of such estimates are lacking and in need for global models. So, I suggest that authors attempt to better constrain and justify their assumptions in their revisions. My decision is to revise extensively and resubmit.

Reviewer #2 (Remarks to the Author):

I enjoyed reading this paper, and I believe that the subject is important and will be of considerable interest to the scientific community (broadly defined). I am not a biogeochemist, and probably can't comment in detail on these aspects of the research, but I am concerned that 1) some of the methods are poorly described, and 2) uncertainly is not adequately quantified. My concerns regarding the methods are outlined in the detailed comments below; in particular, I didn't

understand the approach to routing used by the authors, and I also didn't understand the utility or necessity of the Monte Carlo analyses briefly referred to. Perhaps more importantly, estimates of burial and mineralization are based on a regression equation for primary production with an r^2 of ~ 0.5 and parameters that are also not well-constrained, so the potential for accumulated errors seems significant, and more important, not entirely assessed (I think that the analysis of errors is limited to a sensitivity analysis based on the Monte Carlo sampling of parameter and other variability, but I am kind of guessing about this, and I think the authors should develop this more clearly....so the reader doesn't have to guess).

I have also suggested some relatively minor changes to improve the figures, and other issues elsewhere in the manuscript.

I am hoping that the authors can readily address these issues, and that the manuscript can be move towards eventual publication.

Jim Pizzuto
Dept. of Geological Sciences
University of Delaware

Some detailed comments:

1. line 49. What is a "trend and attribution" analysis? Please avoid jargon – explain this in clear English!
2. Line 58. What measure is used for reservoir size? Surface area? Volume? Also, what is the measure of water discharge? Annual "throughput" or something? Please clarify.
3. Lines 82-83. "although photochemical OC degradation may also contribute". Please explicitly indicate which factors are actually included in the modeling, and which are ignored.
4. Line 109. Should "to" in this sentence actually be "with"?
5. Lines 109-110. "the rising anthropogenic OC loading to rivers". How is this reflected in the model computations? Does this come from Global-NEWS? Please discuss.
6. Paragraph - lines 122-136. I wanted to see the uncertainties in these statistics presented in the text as I read it. Could they be placed in parentheses? I realize that they are included in the figure.
7. Lines 136-137. "90% of all river networks"....What does this statistic actually mean? Does this refer to rivers of any Strahler order? Orders greater than some value? If one dam is built in the Amazon river basin, is this entire network fragmented by a dam? I am not sure that this claim, as stated, really means anything. Please be specific.
8. Line 177. "the increasing OC loading to rivers". This is the second time this is noted in the manuscript. Is there an explanation for this that can be cited? Why is this occurring?
9. Line 239-240. According to my reading of the data, the OC export to the global coastal ocean increased from 1970 to the end of the 20th Century. The dams reduced this increase, as noted by the authors. However, the text might be taken to imply that dams are causing a REDUCTION in OC export to the global coastal ocean overall, which is not the case. Please rewrite and clarify.
10. Figure 2. What is the "stream order" indicated on the x axis. Is this the order of the entire watershed or the order of the watershed upstream of the location of the dam? Can this be clarified?
11. Figure 3. Please indicate the year on each figure, in addition to listing it in the caption. Also add the category as a title at the top of each column of 3 figures: "Respiration" on the left column, and "Burial Flux" on the right. Please provide units for the colorbar on the left.
12. Methods, line 49-50. For each parameter value a pdf is derived from data. Why? Please justify this approach, and explain the goals of using it. It is not obvious.
13. Methods: which of the variables in equation (3) is actually provided by the Global-NEWS database?
14. Methods; line 137. "Global regression relationship were obtained from the Monte Carlo analysis..." Why was this useful? What goals are satisfied by performing this analysis? What IS the

Monte Carlo analysis, anyway? This has not been explained or justified. The Monte Carlo analysis is discussed briefly in the Supplementary Material, but it seems to play an important role in the error analysis, and should be explained in some detail in the methods to help justify that this exercise (i.e., the methods used in the paper) is accurate enough to be useful.

15. Methods: Line 158. "a similar formalism". I have no idea what this means. What is a formalism? "Similar" does not tell the reader what actually was done.

16. Methods: Lines 166-167. Please explain the four "MA" scenarios – I have no idea what these terms are referring to.

17. Methods: equation 10 and related text. I could not follow this "routing scheme" – please explain with greater clarity. Also, Figure S2 does not show a routing procedure – it presents a watershed map with some confusing indices ($k+1$, $k+2$, and k). What is k , anyway? This was not comprehensible to me.

18. Methods: I would like to see a discussion of the uncertainty generated by the use of equations (11), (12) and (13) beyond the sensitivity analysis presented in the Supplementary Material. If the r^2 value of equation (11) is $\sim 1/2$, and this relationship used as a basis for equations (12) and (13) (each of which contains additional uncertainty), it seems as if the predictions have very small precision.

19. Methods, line 227. But equation (4) has an r^2 of only 0.34. See comment 18 above.

20. Methods, lines 231-235. Perhaps more explanation could be provided here. If no other variables other than catchment area are used, this procedure seems to be based on an assumption of constant drainage density....across the entire globe, which seems unwarranted and very inaccurate.

21. Figure S4. The color scheme here is too difficult to differentiate (i.e. the light and dark green and blue shades are too similar). Also, please add a legend to the figure so the reader can understand it without having to read the caption.

Reviewers' comments:

Reviewer #1 (Remarks to the Author):

This manuscript entitled “Global perturbation of organic carbon cycling by river damming” provides a well-structured modelling approach to quantify the impacts of river damming on the global carbon cycle. Although this model is sound in theory, the k -values for the fluxes need to be altered or justified, this is a major flaw as many are not well supported as I will describe below. I understand that the model needs to be simplified, but these are key processes that greatly impact the model and what is learned about this perturbation to the carbon cycle.

We thank the reviewer for his/her feedback. The reviewer is correct that the various rate parameters (k -values) used in the flux equations are central to the proposed modeling approach. Hence, we agree that they must be fully supported. In the revised manuscript we have therefore included additional justifications where appropriate and, in some cases, adjusted how we derive the k -values (as detailed below). We have also revised the general statement at the beginning of the Methods section describing the overall approach we followed to select the flux expressions and the corresponding parameter values.

General Comments:

1. The assumption that POC and DOC have the same reactivity and the assumption that autochthonous OC is 3X more reactive than allochthonous OC need to be further supported with data from the water column. The Gudasz et al (2010) data, used to derive equation 4, includes data from lakes with varying ratios of allochthonous to autochthonous inputs. Therefore, the reactivity of autochthonous and allochthonous OC should be the same in this model if using the Gudasz data.

These are all valid points, which we address in the order of (1) using the Gudasz et al. data for equation 4, (2) rates of POC vs. DOC mineralization, and (3) autochthonous vs. allochthonous mineralization rates. First, however, we wish to point out that published assessments of organic carbon degradation in lentic systems all agree that global mineralization trends exhibit strong dependencies on temperature and hydraulic residence time. These are therefore trends we definitely want to incorporate in how we model mineralization rates. The residence time dependence is inherent to any in-reservoir process because it determines the time scale over which the process operates. In other words, the longer the residence time, the more organic carbon gets mineralized (all other conditions equal). The temperature is built in via equation 4.

(1) We acknowledge that we may have poorly explained the use of the Gudasz et al. data for allochthonous carbon mineralization. Gudasz et al. compiled data on lake sediment organic carbon mineralization rates. In our manuscript we therefore “scaled” the Gudasz et al. carbon mineralization rates in order to match the global allochthonous DOC mineralization rates in lakes compiled by Hanson, et al.¹ through an extensive literature search. We recognize we should have more clearly justified the scaling procedure. We also recognize that the scaling of the Gudasz et al. data introduces an additional step in the parameterization, which we can avoid by directly building on the work of Hanson et al. (which we in fact did in an earlier version of our model).

Thus, in the revised manuscript we now implement the expression for the mineralization rate constant $k_{min,allo}$ for allochthonous DOC, proposed by Hanson, et al.¹, and which Hanson, et al.² tested with water column data from lakes worldwide:

$$k_{min,allo} = k_{20} \times \theta^{(T-20)}$$

where θ is a unitless scaling coefficient equal to 1.07, T is temperature in °C, and k_{20} is the rate constant at 20°C. For the range of k_{20} (which Hanson et al. refer to as R_0) we impose the same range of values Hanson et al. present in their papers (0.256-1.825 yr⁻¹). As we also note in the Supplementary Material, the range of $k_{min,allo}$ for allochthonous DOC used in our model is consistent with the organic carbon mineralization rates for a large variety of freshwater ecosystems along the land-ocean continuum compiled by Catalan, et al.³

The DOC rate constant formulation used in the revised manuscript yields somewhat lower values than those obtained with the formulation in our original submission. Upon closer inspection, the new range of values is more realistic than the previous one: together with our revised estimate of the relative reactivity of autochthonous POC and DOC (see below), our results now agree very well with the overall rates of organic carbon mineralization in Catalan et al.'s compilation.

(2) We agree that the assumption regarding identical rate constants of mineralization of allochthonous POC and DOC needs better justification. A review of the existing literature reveals that the ratios of the mineralization rate constants of POC to DOC vary substantially: in some systems DOC mineralization rate constants exceed that of POC⁴, while the opposite is found in others⁴⁻⁶. Additionally, many studies only report a “lumped” POC degradation rate, that is, allochthonous plus autochthonous POC, which in many cases is “anomalously” high due to admixing of more reactive autochthonous material^{6,7}. Thus, given the current state of knowledge, it is impossible to unambiguously decide whether, on average, the degradation rate constant of allochthonous POC is larger, smaller or equal to that of coexisting DOC. In view of this, we opted for the simplest assumption: equal rate constants.

Upon reflection, the latter assumption is equally difficult to justify. We therefore return to an earlier model formulation, where we used the same probability density functions (PDF) to calculate the DOC and POC mineralization rate constants, but allowed these parameters to vary independently in the Monte Carlo analysis. This approach therefore generates ratios of allochthonous POC and DOC mineralization rate constants that are smaller or larger than one. We have revised the text (methods lines 105-118) and model results accordingly and added the references above to justify the approach. We further acknowledge that there remains a knowledge gap about the mineralization of autochthonous POC in reservoirs.

The allochthonous POC mineralization flux equations generated by the Monte Carlo analysis using the approach outlined above do not differ significantly from those of the original manuscript. However, as expected, the increased variability reduces the R² values of the fitted global equations (Table S2) and increases the overall uncertainties on the model outcomes (see lines 48-57 in the supplementary material).

(3) It is generally agreed that, on average, in-reservoir produced autochthonous organic matter tends to be more labile than allochthonous organic matter. We therefore tested a range of scaling factors and found that a 3-times higher autochthonous reactivity yielded the best fit of our model-predicted total carbon mineralization rates to the rates compiled by Catalan, et al.³ across a wide range of aquatic ecosystems and climate zones.

While in the original manuscript the scaling factor was kept constant at a value of 3, we now use a normal distribution PDF with a mean value of 3 and a range between 1 and 6. With this range of scaling factors our model generates carbon mineralization rates that encompass the entire Catalan,

et al.³ data set (with the exception of 4 outliers, 3 of which are from the same small stream). We show this in the supplementary material with the addition of the following figure:

In this figure, the first 6 boxes represent the values output by the Monte Carlo analysis under various constraints. The first three boxes are the constraints used in the final model, i.e. the allochthonous POC and DOC distributions for k_{\min} , and the autochthonous k_{\min} that uses a normal distribution for the k_{20} scaling factor, with a mean of 3 (Table S2). The boxes labelled “Auto, x3”, “Auto, x1”, and “Auto, x6” show additional Monte Carlo outputs where we assume relative autochthonous reactivity is fixed as 3 times more reactive, equally reactive, and 6 times more reactive. The last box shows the distribution of lumped POC, DOC, and allochthonous and autochthonous values presented by Catalan et al. We see that the Catalan data is well reproduced by the constraints we have fixed in the model in its latest iteration, while increasing or decreasing the relative reactivity of autochthonous material produces data that does not adequately reproduce the Catalan trends. We have added this figure to the supplementary material as Figure S7.

With this change and the others described above, there is now a $\pm 33\%$ (parameter) uncertainty associated with the autochthonous carbon mineralization fluxes, compared to $\pm 28\%$ previously.

All the above changes are now included in the revised submission.

2. Can the authors provide more evidence to support excluding photochemistry in the model?

This was an unfortunate formulation: what we really wanted to say is that we do not explicitly distinguish between photochemical and microbial mineralization. The mineralization rate constant ranges we use are based on the work of Hanson, et al.^{1,2} (see above response to comment #1), who define “mineralization at the ecosystem scale, R_E .” In their 2011 paper¹, they explicitly define R (i.e. R_0 or R_E) as “respiration plus photo-oxidation.” In other words, in our model mineralization includes photochemical mineralization. We have revised the text (lines 82-83) accordingly.

3. It would be useful if the fluxes were reported in terragrams instead of terramoles for comparison with other studies.

We have included values in Tg C in brackets after all molar values in the revised manuscript.

4. Can the authors account for riverine loss of OC during transport between reservoirs?

In-river OC losses are accounted for by using the Global-NEWS POC and DOC yields. The Global-NEWS yields are calibrated using measured DOC and POC export fluxes at the river mouths. For an undammed river basin, the export fluxes correspond to the gross yields to the river minus the riverine losses. We used the Global-NEWS DOC and POC yields to calculate the input fluxes to a reservoir and then corrected these fluxes iteratively for the in-reservoir processes until the river export fluxes to the coast matched those originally reported in Global-NEWS. As the Global-NEWS DOC and POC yields are estimated per river basin, our simulation results are valid at the scale of entire basins or higher. We have revised the methods section to include this information (lines 40-42).

5. Can the authors test the model on a specific reservoir or river that is well studied to further ground-truth the model?

As stated in our response to comment #4, the resolution of our model does not extend below the catchment scale. Therefore, any model validation attempt using single reservoirs has limited statistical meaning. Even the combination of all reservoir OC budgets that can be found in the literature does not yield a statistically representative sample of reservoirs worldwide. It is therefore impossible to know with certainty whether trends observed for these existing budgets can be extrapolated to the entire global population of reservoirs.

The best candidate to crosscheck the model output is a large river basin. Fortunately, Li et al.⁸ recently estimated the reduction of organic carbon export by the Yangtze River due to the construction of dams. Their estimate is based on analysing sediment samples retrieved in the lower part of the river. The approach is thus completely independent from our model calculations. Li et al.'s estimate a reduction of $408 \pm 158 \text{ Gmol C yr}^{-1}$ after closure of the Three Gorges Dam (2006), when compared to the 1950s (i.e., pre-dam conditions). This is actually quite close to our estimate of 324 Gmol yr^{-1} in year 2030, considering the uncertainties associated with our and Li et al.'s estimation methods. We now include this comparison in the text (main text, lines 217-222, and supplementary material, lines 107-117).

All of these issues except number 3 are serious problems that need justification and elaboration before the model can be fully evaluated.

Specific Comments:

Line 111, 115: The fluxes presented in text for 1970 and 2000 do not match the fluxes in Figure 1.

This mistake has been rectified. All in-text numerical values now agree with the figures, and reflect the revised model results.

Methods Line 227: Could the author provide evidence/references for the use of equation 4 for calculating the k value for TOC decay in flooded soils?

For consistency, we use the same Hanson et al. temperature function (i.e., $\theta^{(T-20)}$) for all organic carbon pools (see above). For the flooded soils, we adjusted the pre-function coefficient so as to reproduce the decay time scale of flooded soil organic matter typically observed after dam closure (5-10 years). This has been clarified in the methods section, lines 267-287.

In conclusion, I think that the importance of such estimates are lacking and in need for global models. So, I suggest that authors attempt to better constrain and justify their assumptions in their revisions. My decision is to revise extensively and resubmit.

We have followed the reviewer's advice and agree that it helped us to better constrain and justify the assumptions and modeling results.

Reviewer #2 (Remarks to the Author):

I enjoyed reading this paper, and I believe that the subject is important and will be of considerable interest to the scientific community (broadly defined). I am not a biogeochemist, and probably can't comment in detail on these aspects of the research, but I am concerned that 1) some of the methods are poorly described, and 2) uncertainty is not adequately quantified. My concerns regarding the methods are outlined in the detailed comments below; in particular, I didn't understand the approach to routing used by the authors, and I also didn't understand the utility or necessity of the Monte Carlo analyses briefly referred to. Perhaps more importantly, estimates of burial and mineralization are based on a regression equation for primary production with an r^2 of ~ 0.5 and parameters that are also not well-constrained, so the potential for accumulated errors seems significant, and more important, not entirely assessed (I think that the analysis of errors is limited to a sensitivity analysis based on the Monte Carlo sampling of parameter and other variability, but I am kind of guessing about this, and I think the authors should develop this more clearly....so the reader doesn't have to guess).

The reviewer's comments have prompted us to clarify our approach and methods in the revised manuscript. We have previously used the same general modelling approach to predict nutrient silicon and phosphorus retention in reservoirs^{8,9}, which we cite in the manuscript. Those papers provide a detailed justification of the approach taken. Therefore, while we now provide a more in-depth description of our approach and assumptions, we also specifically refer to our earlier work where appropriate. In the revised manuscript, we include an introductory paragraph in the methods section that summarizes the general approach. We have updated the methods throughout to more clearly explain why each specific step is taken. Below, we succinctly explain the value of the modelling approach, the Monte Carlo approach, and the Global-NEWS model, including its benefits, and the Millennium Ecosystem Assessment (MA) scenarios.

Existing approaches for making global scale estimates of the effects of damming on carbon and nutrient riverine fluxes have relied on empirical approaches¹⁰⁻¹³. Typically, these studies extrapolate the "average" fixation, burial and mineralization fluxes observed in a limited number of reservoirs to the entire world. While this approach is useful to answer certain preliminary research questions, it cannot be validated statistically. Attempting to draw any sort of global conclusions from empirical relationships derived from, say, 20-100 reservoirs, usually clustered in Europe and North America, is fraught with unconstrained uncertainties. There is simply no way of checking whether the reservoirs for which data exists actually fit the global trends, or if they are outliers.

To overcome this difficulty, we have developed a new mechanistic modelling approach, which the authors (TM and PVC) previously applied to predict global reservoir silicon and phosphorus retention^{8,9}. We use a similar mass balance approach to model in-reservoir organic carbon fluxes and transformations (the arrows in Figure S1). The approach allows us to incorporate the large amount of data and knowledge about organic carbon cycling in aquatic environments into the model. Mathematically, the data (and knowledge) are translated into probability density functions that account for the mean values and spread of the model parameters controlling the fate of

organic carbon in reservoirs. As a simple example, though reservoir volumes range from 0.001 to 180 km³ worldwide, small reservoirs are exponentially more common than large ones, and so a Pareto distribution is used to account for this trend. Next, we perform a Monte Carlo analysis, which means that we run the model over and over again, each time with a set of parameter values chosen randomly from the assigned probability density functions. In the present case, we do this 6000 times and obtain distributions for each of the fluxes shown on Figure S1. From this dataset of model-generated fluxes, we derive relationships between carbon fluxes and reservoir water residence times, which we then apply individually to each of the 6848 reservoirs included in the global GRanD database of existing reservoirs, and for the 2030 and 2050 estimates, the new database on dams under construction or expected to be built by 2030.

One additional piece of information we need to apply the above approach are the inflow fluxes of allochthonous DOC and POC into each reservoir (at each of the chosen time points). This is where we use the output of the Global-NEWS model, which is currently the only spatially explicit model that yields estimates of DOC and POC loads in watersheds. By spatially overlaying the reservoirs in GRanD with the Global-NEWS output, we are able to estimate the influxes of POC and DOC to each reservoir based on the upstream watershed area and the amount of OC leaving the dams upstream. For the 2030 scenarios, the GRanD database is expanded to include the dams currently under construction or expected to be built by 2030.

Global-NEWS is particularly useful in that, in addition to providing DOC and POC loadings in year 2000, it also allows to hindcast to year 1970, and project to 2030 and 2050 using the Millennium Ecosystem Assessment (MA) scenarios developed by the United Nations. These scenarios can be used in conjunction with our model to estimate how the current boom in dam construction will affect global nutrient and carbon fluxes in the coming decades.

Global-NEWS predicts nutrient and carbon loads to watersheds based on a series of relationships between loading and parameters associated with land use (e.g. agriculture, wetland, forested), geomorphology (e.g. slope), soil type, and climate (e.g. precipitation and temperature). The various processes accounted for are summarized in the figure shown below¹². For the different time points and scenarios, the predicted nutrient and carbon loads mainly diverge because of variations in land use and climate. The MA scenarios predict a world that either becomes more globalized (Global Orchestration and TechnoGarden) or regionalized (Adapting Mosaic and Order from Strength), and where society takes either a proactive approach to environmental management and governance (TechnoGarden and Adapting Mosaic) or a reactive approach (Global Orchestration and Order from Strength). So for example, the Global-NEWS model predicts that for the Global Orchestration (GO) scenario, there is a massive shift towards agricultural production in Asia, which results in increased phosphorus loads to watersheds via fertilizer application. In our model, the GO scenario therefore yields greater primary productivity in reservoirs than for the other scenarios. (For details regarding how each of the scenarios were translated into Global-NEWS model inputs, see Seitzinger, et al. ¹⁴)

I have also suggested some relatively minor changes to improve the figures, and other issues elsewhere in the manuscript.

I am hoping that the authors can readily address these issues, and that the manuscript can be move towards eventual publication.

Jim Pizzuto
 Dept. of Geological Sciences
 University of Delaware

Some detailed comments:

1. line 49. What is a “trend and attribution” analysis? Please avoid jargon – explain this in clear English!

We have changed this sentence to:

“In this study, we present a worldwide analysis of decadal trends in riverine OC fluxes that explicitly accounts for reservoirs as a dynamic compartment of the C cycle on the continents.”

2. Line 58. What measure is used for reservoir size? Surface area? Volume? Also, what is the measure of water discharge? Annual “throughput” or something? Please clarify.

We have replaced “size” by “volume.” Discharge indeed refers to the flow through the dam. We have therefore changed the text to read, “discharge through the dam”.

3. Lines 82-83. “although photochemical OC degradation may also contribute”. Please explicitly indicate which factors are actually included in the modeling, and which are ignored.

Photochemistry is included (see response to reviewer 1).

4. Line 109. Should “to” in this sentence actually be “with”?

Yes, it has been changed.

5. Lines 109-110. “the rising anthropogenic OC loading to rivers”. How is this reflected in the model computations? Does this come from Global-NEWS? Please discuss.

Yes, changes in OC loads are obtained from Global-NEWS. Please see detailed explanation to the general comment by the reviewer. We have changed the text to elaborate on the Global-NEWS model load predictions, the MA scenarios, and how these serve as input to our model calculations (methods, lines 40-48).

6. Paragraph - lines 122-136. I wanted to see the uncertainties in these statistics presented in the text as I read it. Could they be placed in parentheses? I realize that they are included in the figure.

We have added the uncertainties in the text throughout.

7. Lines 136-137. “90% of all river networks”....What does this statistic actually mean? Does this refer to rivers of any Strahler order? Orders greater than some value? If one dam is built in the Amazon river basin, is this entire network fragmented by a dam? I am not sure that this claim, as stated, really means anything. Please be specific.

We are referring to the degree of river volume fragmentation, according to the River Fragmentation Index (RFI), presented in Grill, et al.¹⁵ A river with no fragmentation will have an RFI of 0%, up to a maximum of 100%. A dam that splits a river into two equal volume portions will have an RFI of 50%. We have revised the text so it read, “Although it is unlikely that damming activity will cease abruptly, a slowing down in dam construction can be expected by 2030, at which time fragmentation by dams will affect more than 90% of the total river volume on Earth¹⁵.”

8. Line 177. “the increasing OC loading to rivers”. This is the second time this is noted in the manuscript. Is there an explanation for this that can be cited? Why is this occurring?

Increasing OC loading to rivers occurs primarily due to land use changes, particularly deforestation, and climate change¹⁶. With more bare soil exposed, organic matter is subjected to increased erosion during rain events, as well as from wind, and these loads eventually end up in rivers, thus increasing the global OC load carried by rivers. Other anthropogenic influences such as mining and quarrying similarly disturb surface soils and lead to the mobilization of organic matter to waterways. We have added citations of global-scale studies that predict this, and added a brief explanation for why this is occurring (181-183).

9. Line 239-240. According to my reading of the data, the OC export to the global coastal ocean increased from 1970 to the end of the 20th Century. The dams reduced this increase, as noted by

the authors. However, the text might be taken to imply that dams are causing a REDUCTION in OC export to the global coastal ocean overall, which is not the case. Please rewrite and clarify.

We acknowledge that the text may be ambiguous and have rephrased the paragraph to read:

“According to our calculations, in 2000, dams lowered the OC export to the global coastal zone via rivers by 13%: 7% through sequestration of OC in reservoir sediments and 6% through in-reservoir mineralization (Table S6). For comparison, for the same period, dams decreased riverine export of reactive silicon and phosphorus by only 5 and 12%, respectively^{8,9}. All other factors equal, by eliminating OC more efficiently than nutrients, river damming should enhance carbon fixation over mineralization in the receiving waters. With the ongoing rise in the number of dams, riverine OC export in 2030 could be 19% lower than if no dams were built after 2000 (12% due to burial and 7% to mineralization). Therefore, in all but one of the MA scenarios, the current boom in dam construction could even causes a net decrease in OC export to the coastal zone between 2000 and 2030, despite the increased anthropogenic loading of OC to watersheds (Table S6). Through its large impact on the delivery of riverine OC, damming therefore represents a major anthropogenic forcing on the trophic conditions and carbon balance of the coastal ocean.”

10. Figure 2. What is the “stream order” indicated on the x axis. Is this the order of the entire watershed or the order of the watershed upstream of the location of the dam? Can this be clarified?

We have changed the figure caption to:

“Total OC burial (autochthonous + allochthonous) in dam reservoirs as a function of Strahler stream order for the global river network. The 2030 and 2050 OC burial estimates correspond to the Global Orchestration (GO) Millennium Assessment (MA) scenario. Inset: distribution of dams based the Strahler stream order on which they are located. Note that the number of dams in 2050 is assumed to be equal to that in 2030. See Table S3 for the assignment of Strahler stream orders.”

11. Figure 3. Please indicate the year on each figure, in addition to listing it in the caption. Also add the category as a title at the top of each column of 3 figures: “Respiration” on the left column, and “Burial Flux” on the right. Please provide units for the colorbar on the left.

All these suggested changes have been made to Figure 3.

12. Methods, line 49-50. For each parameter value a pdf is derived from data. Why? Please justify this approach, and explain the goals of using it. It is not obvious.

Please refer to our detailed explanation to the reviewer’s general comment.

13. Methods: which of the variables in equation (3) is actually provided by the Global-NEWS database?

None of the variables in equation 3 are provided by Global-NEWS. (Note: Global-NEWS is a model, not a database.) We clearly state in the paragraph that introduces equation 3 that the equation is generated using variables from the GRanD database.

14. Methods; line 137. “Global regression relationship were obtained from the Monte Carlo

analysis...” Why was this useful? What goals are satisfied by performing this analysis? What IS the Monte Carlo analysis, anyway? This has not been explained or justified. The Monte Carlo analysis is discussed briefly in the Supplementary Material, but it seems to play an important role in the error analysis, and should be explained in some detail in the methods to help justify that this exercise (i.e., the methods used in the paper) is accurate enough to be useful.

As with reviewer 2’s comment #12, please refer to our response to his general comment.

15. Methods: Line 158. “a similar formalism”. I have no idea what this means. What is a formalism? “Similar” does not tell the reader what actually was done.

We mean that the same equation, rearranged slightly, is the same as the one used by Vollenweider in his landmark 1976 paper to describe phosphorus burial in lakes. This makes sense as Vollenweider used a mass balance approach very similar to the one we use in our model. We use the “similar” because OC removal is due to both mineralization and burial, whereas Vollenweider’s loss term only accounted for burial. We state this explicitly in the second half of the sentence. For clarity, we have changed this sentence to:

“Note that Eqs. (8) and Eq. (9) are formally similar to the equation derived by Vollenweider¹⁷ to describe phosphorus cycling in lakes, though scaled with a_i and b_i to account for the separation of loss fluxes into mineralization and burial, rather than lumped together as in Vollenweider’s derivation.”

16. Methods: Lines 166-167. Please explain the four “MA” scenarios – I have no idea what these terms are referring to.

“MA” refers to the Millennium Ecosystem Assessment scenarios developed by the United Nations. We have also added an explanation of these scenarios to the methods, lines 44-48.

17. Methods: equation 10 and related text. I could not follow this “routing scheme” – please explain with greater clarity. Also, Figure S2 does not show a routing procedure – it presents a watershed map with some confusing indices ($k+1$, $k+2$, and k). What is k , anyway? This was not comprehensible to me.

We acknowledge that Figure S2 does not in itself show the routing procedure; it is meant to help explain equation 10 and the variables included. We have changed the figure caption to:

“Schematic representation of the breakdown of a hypothetical watershed into the sub-watersheds that are hydrologically connected to the dam reservoirs in the watershed. The figure helps explain the routing procedure described in Methods section 4 and equation 10.”

We have modified the k notation slightly: please note that k refers to a given reservoir for which burial and mineralization fluxes are calculated in the routing scheme, $k-1$ is the first reservoir upstream, $k-2$ is the second reservoir upstream, and so on. This is explained in the text following equation 10, where we define the variables.

We have modified the text with an additional sentence at the end of the paragraph that explains equation 10:

“All reservoirs are spatially routed to account for cascading dams and dams on tributaries (Figure S2). The total influx (mol yr⁻¹) of allochthonous POC or DOC to a reservoir k is then given by:

$$F_{k,in} = \sum_1^n F_{k-1,out} + (W_k - \sum_1^n W_{k-1}) \times Y_k \quad (10)$$

where $\sum_1^n F_{k-1,out}$ is the sum of the fluxes of allochthonous POC or DOC leaving all dams immediately upstream of reservoir k on tributaries 1 to n , W_k is the total upstream watershed area (km²), $\sum_1^n W_{k-1}$ is the sum of the upstream watershed areas of dams $k-1$, and Y_k is the POC or DOC yield (mol km⁻² yr⁻¹) in the undammed, $(W_k - \sum_1^n W_{k-1})$, watershed area downstream of dams $k-1$. In other words, for each dam, the OC load leaving all upstream dams is added to the load mobilized from the (undammed) watershed area directly flowing into the reservoir.”

18. Methods: I would like to see a discussion of the uncertainty generated by the use of equations (11), (12) and (13) beyond the sensitivity analysis presented in the Supplementary Material. If the r^2 value of equation (11) is $\sim 1/2$, and this relationship used as a basis for equations (12) and (13) (each of which contains additional uncertainty), it seems as if the predictions have very small precision.

It would appear that the reviewer is unclear on how we use the Monte Carlo analyses as the basis for estimating uncertainties. Below, we explain our approach again. We have also revised the text so as to clarify our procedure. We hope that the reviewer (and reader) will appreciate that, given the current state of knowledge and data availability, our approach provides the best means to quantify uncertainties on global scale estimates of the effects of dams on riverine OC fluxes.

Our first run of Monte Carlo simulations generates 6000 reservoir OC budgets that are consistent with our model structure and parameter ranges. These 6000 simulations yield the burial and mineralization versus water residence time relationships of Figure S3 (reproduced below). For each of the binned categories, we present the median, standard deviations, and 25% and 75% confidence intervals, plus outliers. Each of these categories is fitted to a separate gamma distribution, which “summarizes” the probability of achieving a certain mineralization or burial efficiency based on the reservoir’s water residence time. The gamma distributions therefore reflect the combined effect of all the parameters ranges and distributions that we have built into our mechanistic model (which are shown in Table S1). Next, we apply these gamma distributions to all the reservoirs in the global database, and run a second Monte Carlo analysis using the gamma functions. This second Monte Carlo analysis randomly selects burial and mineralization values from the gamma distributions based on the water residence time of each individual GRanD reservoir. This approach thus approximates the range of possible values that could reasonably be predicted for a given water residence time. By running this approximation over and over for all reservoirs in GRanD, we thus predict the overall uncertainty associated with the original (i.e. 1st) Monte Carlo output. Our analysis therefore produces a lumped assessment of the uncertainty associated with the model parameters when applied to a real dataset.

A key aspect of our model approach is that as we increase the number of dams and reservoirs included in the analysis the uncertainty on the global estimations actually decrease (in other words, the more reservoirs we include, the more the combined results will approach the mean behaviour). For example, our analysis shows that k_{bur} is a very sensitive model parameter, the associated uncertainty, once applied to a global dataset, is actually quite reasonable. In a similar vein, even moderate modifications of equations 12 and 13 result in relatively small changes to the actual magnitudes of OC buried or mineralized in reservoirs, on the order of $\pm 18\%$.

The reviewer is correct in noting an R^2 of 0.5 for the photosynthesis equation may seem low but, again, through the bootstrap analysis we describe in the second paragraph of the supplementary material, we show that once applied at the global scale, the associated uncertainty is only on the order of $\pm 10\%$.

19. Methods, line 227. But equation (4) has an r^2 of only 0.34. See comment 18 above.

See response to comment 18 above.

20. Methods, lines 231-235. Perhaps more explanation could be provided here. If no other variables other than catchment area are used, this procedure seems to be based on an assumption of constant drainage density....across the entire globe, which seems unwarranted and very inaccurate.

The validity of our approach is supported by the R^2 values for the correlation between actual (i.e., observed) stream orders for European ($N = 2192$) and South American dams ($N = 1602$)¹⁸ and the stream order calculated from the upstream catchment area. For Europe, $R^2 = 0.90$, and for South America, $R^2 = 0.87$. Given the high degrees of correlation, and the fact that Europe and South America represent more than a third of the dams in 2030 and 2050, we are confident that the catchment areas used in the global-scale calculations are reasonable. We have added this information to the methods, lines 293-297.

21. Figure S4. The color scheme here is too difficult to differentiate (i.e. the light and dark green and blue shades are too similar). Also, please add a legend to the figure so the reader can understand it without having to read the caption.

We have changed the colour scheme and added a legend.

References:

- 1 Hanson, P. C. *et al.* Fate of allochthonous dissolved organic carbon in lakes: a quantitative approach. *PloS one* **6**, e21884 (2011).
- 2 Hanson, P. C., Buffam, I., Rusak, J. A., Stanley, E. H. & Watras, C. Quantifying lake allochthonous organic carbon budgets using a simple equilibrium model. *Limnology and Oceanography* **59**, 167-181 (2014).
- 3 Catalan, N., Marce, R., Kothawala, D. N. & Tranvik, L. J. Organic carbon decomposition rates controlled by water retention time across inland waters. *Nature Geosci* **9**, 501-504 (2016).
- 4 Ostapenia, A. P., Parparov, A. & Berman, T. Lability of organic carbon in lakes of different trophic status. *Freshwater biology* **54**, 1312-1323 (2009).
- 5 Wetzel, R. G. Detrital dissolved and particulate organic carbon functions in aquatic ecosystems. *Bulletin of Marine Science* **35**, 503-509 (1984).
- 6 Hanson, P. C. *et al.* A model of carbon evasion and sedimentation in temperate lakes. *Global Change Biology* **10**, 1285-1298 (2004).
- 7 Cole, J. J., Carpenter, S. R., Kitchell, J. F. & Pace, M. L. Pathways of organic carbon utilization in small lakes: Results from a whole - lake ¹³C addition and coupled model. *Limnology and Oceanography* **47**, 1664-1675 (2002).
- 8 Maavara, T., Dürr, H. H. & Van Cappellen, P. Worldwide retention of nutrient silicon by river damming: From sparse data set to global estimate. *Global Biogeochemical Cycles* **28**, 842-855, doi:10.1002/2014GB004875 (2014).
- 9 Maavara, T. *et al.* Global phosphorus retention by river damming. *Proc Natl Acad Sci U.S.A.* **112**, 15603-15608 (2015).
- 10 Harrison, J. A., Frings, P. J., Beusen, A. H. W., Conley, D. J. & McCrackin, M. L. Global importance, patterns, and controls of dissolved silica retention in lakes and reservoirs. *Global Biogeochemical Cycles* **26**, 12, doi:10.1029/2011gb004228 (2012).
- 11 Harrison, J. A. *et al.* The regional and global significance of nitrogen removal in lakes and reservoirs. *Biogeochemistry* **93**, 143-157 (2009).
- 12 Mayorga, E. *et al.* Global Nutrient Export from WaterSheds 2 (NEWS 2): Model development and implementation. *Environ. Modell. Softw.* **25**, 837-853, doi:10.1016/j.envsoft.2010.01.007 (2010).
- 13 Dean, W. E. & Gorham, E. Magnitude and significance of carbon burial in lakes, reservoirs, and peatlands. *Geology* **26**, 535-538 (1998).
- 14 Seitzinger, S. P. *et al.* Global river nutrient export: A scenario analysis of past and future trends. *Global Biogeochemical Cycles* **24**, GB0A08, doi:10.1029/2009gb003587 (2010).
- 15 Grill, G. *et al.* An index-based framework for assessing patterns and trends in river fragmentation and flow regulation by global dams at multiple scales. *Environmental Research Letters* **10**, 015001 (2015).
- 16 Regnier, P. *et al.* Anthropogenic perturbation of the carbon fluxes from land to ocean. *Nat. Geosci.* **6**, 597-607, doi:10.1038/ngeo1830 (2013).

- 17 Vollenweider, R. A. Input-output models with special reference to the phosphorus loading concept in limnology. *Schweizerische Zeitschrift für Hydrologie* **37**, 53-84 (1975).
- 18 Lehner, B., Verdin, K. & Jarvis, A. New global hydrography derived from spaceborne elevation data. *Eos* **89**, 93-94 (2008).

Reviewers' comments:

Reviewer #1 (Remarks to the Author):

I have now examined the extensive revisions made by the authors' and feel that the manuscript is now ready for publication. They have addressed all of my comments adequately. In particular, I am very satisfied with the new way, via Monte Carlo, they use to derive the POC and DOC decay rates. I think this is much better even though the r^2 values have dropped with the greater variability. All other minor edits were also satisfactorily dealt with. I think this is a very timely publication that offers for the first time, a modeling approach to better understand the effects of dam reservoirs on OC cycling in river catchments around the world, something sorely needed and currently missing in the literature.

Reviewer #3 (Remarks to the Author):

NCOMMS-16-18311A

Global perturbation of organic carbon by river damming
Reviewed by Katherine Skalak, USGS

Key points (major claims of the paper)

In reservoirs, $P < R$ overall, but the P/R ratio varies significantly. At the beginning of the 21st century river damming resulted in reservoir burial and mineralization of about 13% of organic carbon carried by rivers to the ocean and this is projected to rise to 19% by 2030. There is a decrease in particulate organic carbon observed in many rivers after closure, but global changes are poorly quantified and photosynthetic carbon fixation and mineralization are highly uncertain. This paper contributes a worldwide analysis of decadal trends in riverine OC fluxes that accounts for reservoirs, predict P and R between 1970 and 2050.

They use the GRanD database to get volume data, etc. for calculations. Assumptions – no more construction after 2030, database that exists on current construction is completed after 2030, P is limiting nutrient for photosynthesis.

The input fluxes of allochthonous POC and DOC supplied by the upland catchment area are calculated from the corresponding watershed yields of Global NEWS model for 1970 and 2000.

General comments:

I think this is an important and novel publication, very worthy of Nature Communications. I think it fills a critical knowledge gap in global carbon cycling which will improve predictions from climate change modeling and global CO_2 cycling. It is well-written and concise. I am suggesting some relatively minor edits to provide clarity to a broad readership.

The statistical approach is sound and the level of detail provided appears sufficient for a researcher to reproduce the work, given the level of detail provided. However, I think it will improve the readability and ability for a reader to understand the full scope of the work if additional details are provided within the paper itself (in methods or SI) rather than only being referenced. Specifics are noted below.

I think the caveats of the approach need to be addressed and the potential consequences for these caveats. For example, there are processes that are not directly addressed which deliver carbon to the landscape and that is either a result of them not being included in the analysis or they are included but not mentioned as output from Global-NEWS.

The critical processes that struck me when reading the paper are: What about carbon associated with geomorphic processes? Carbon associated with sediment from bank erosion and upland delivery of forested soils? How are the carbon loads determined? Is this particulate carbon

associated with sediment or something else? What about large wood and CPOM? What about the role of floods in reservoirs which can introduce substantial large wood and sediment? What about reservoir infilling? This is perhaps not an issue on the largest reservoirs but could be important to numerous mid-sized reservoirs over the course of the model timeframe. Here is just one example:

Zhang, Qian, Robert M. Hirsch, and William P. Ball. "Long-term changes in sediment and nutrient delivery from Conowingo dam to Chesapeake Bay: effects of reservoir sedimentation." *Environmental science & technology* 50, no. 4 (2016): 1877-1886..

I think many of these questions could be addressed with a concise presentation of the output of Global-NEWS within the methods or Supplementary Information. I also think a brief consideration of the scenarios within Global-NEWS should be provided to a reader who is not familiar.

Line 29 – are all references to OC assumed to be TOC?

Line 29 to 31 – these two sentences are confusing as they are written.

Line 44 – specify within the reservoir region.

Lines 87 to 89 – This is not entirely clear as it is written.

Line 93 – this needs a bit of clarifying. What comes out via dam outflow? Are you assuming 100% trapping of POC and release of DOC? Figure 1 is not completely clear and refers to TOC coming out downstream. How do you know what is released from one reservoir and into the next? Or is this process not considered explicitly and you just account for the input downstream from the next set of Global-NEWS output (according to Figure S2).

Line 126 - why does the HRT of reservoirs decrease after this time? Are they smaller or built differently to reduce the HRT?

Line 183 – Is it possible that you are overlooking key processes of delivery? This is difficult to assess in the current version. Caveats of the modeling approach should be presented somewhere.

Figure 1. Cascades of dams – I am having trouble interpreting the cascades of dams. This assumes that DOC is passing through a reservoir, but that POC is not, correct? Based on the schematic, I think the reader might get the incorrect assumption that the output is something coming from out of a dam rather than the budget for all reservoirs globally. It looks like a single reservoir pictured in the background and that might be misleading. This isn't really the output from a dam, but what reaches the ocean from all dams, right? Are there any consequences for the carbon if it passes through multiple dams?

Supplementary Information

Figure S1 – I think it might be more useful for this figure to be more process based if possible, showing a water column flowing into a reservoir and linking various processes through arrows. Or consider a budget approach where inputs are shown as positive contributions and outputs are considered negative. The caption needs clarity. What are the different arrow colors representing?

Figure S2 – caption needs clarity. K1 a model node? A dam? Although it is defined in the text, it should be included in the caption.

Figure S7 caption – Line 245 – this should be outliers, correct? See comment regarding discussion of this figure in the text.

Methods

Line 24 – how is success established? What is the criteria for successful prediction?

Line 89 – Why? These are riverine fluxes – I understand you are dealing with reservoirs that might mimic lentic processes, but do you have evidence to suggest this is similar for riverine fluxes?

Line 129 – This figure does not show the outliers. The way it is referred to suggests that it might.

Line 179 to Line 181 – I think this needs to be more specific, especially with respect to sedimentation rates. Additionally, some of the references are dated.

Line 215 – Is there any way to provide additional information about these scenarios for a reader who is unfamiliar?

Line 229 – cascading dams change to dams that occur longitudinally in series to be more specific if

possible.

Skalak, Katherine J., Adam J. Benthem, Edward R. Schenk, Cliff R. Hupp, Joel M. Galloway, Rochelle A. Nustad, and Gregg J. Wiche. "Large dams and alluvial rivers in the Anthropocene: The impacts of the Garrison and Oahe Dams on the Upper Missouri River." *Anthropocene* 2 (2013): 51-64.

Line 247 – an initial hypothesis...

Reviewers' comments:

Reviewer #1 (Remarks to the Author):

I have now examined the extensive revisions made by the authors' and feel that the manuscript is now ready for publication. They have addressed all of my comments adequately. In particular, I am very satisfied with the new way, via Monte Carlo, they use to derive the POC and DOC decay rates. I think this is much better even though the r^2 values have dropped with the greater variability. All other minor edits were also satisfactorily dealt with. I think this is a very timely publication that offers for the first time, a modeling approach to better understand the effects of dam reservoirs on OC cycling in river catchments around the world, something sorely needed and currently missing in the literature.

We appreciate the time and effort Reviewer 1 invested in reviewing our manuscript a second time and thank him/her for his/her insightful comments.

Reviewer #3 (Remarks to the Author):

NCOMMS-16-18311A

Global perturbation of organic carbon by river damming
Reviewed by Katherine Skalak, USGS

Key points (major claims of the paper)

In reservoirs, $P < R$ overall, but the P/R ratio varies significantly. At the beginning of the 21st century river damming resulted in reservoir burial and mineralization of about 13% of organic carbon carried by rivers to the ocean and this is projected to rise to 19% by 2030. There is a decrease in particulate organic carbon observed in many rivers after closure, but global changes are poorly quantified and photosynthetic carbon fixation and mineralization are highly uncertain. This paper contributes a worldwide analysis of decadal trends in riverine OC fluxes that accounts for reservoirs, predict P and R between 1970 and 2050.

They use the GRanD database to get volume data, etc. for calculations. Assumptions – no more construction after 2030, database that exists on current construction is completed after 2030, P is limiting nutrient for photosynthesis.

The input fluxes of allochthonous POC and DOC supplied by the upland catchment area are calculated from the corresponding watershed yields of Global NEWS model for 1970 and 2000.

General comments:

I think this is an important and novel publication, very worthy of Nature Communications. I think it fills a critical knowledge gap in global carbon cycling which will improve predictions from climate change modeling and global CO_2 cycling. It is well-written and concise. I am suggesting some relatively minor edits to provide clarity to a broad readership.

We acknowledge the reviewer's positive evaluation and greatly appreciate her suggestions to improve the manuscript.

The statistical approach is sound and the level of detail provided appears sufficient for a researcher to reproduce the work, given the level of detail provided. However, I think it will improve the readability and ability for a reader to understand the full scope of the work if additional details are provided within the paper itself (in methods or SI) rather than only being referenced. Specifics are noted below.

We agree that moving additional information into the main text or methods strengthens the manuscript. We have followed the suggestions of the reviewer but, because of the word limits of the journal, we have had to make choices.

I think the caveats of the approach need to be addressed and the potential consequences for these caveats. For example, there are processes that are not directly addressed which deliver carbon to the landscape and that is either a result of them not being included in the analysis or they are included but not mentioned as output from Global-NEWS.

The critical processes that struck me when reading the paper are: What about carbon associated with geomorphic processes? Carbon associated with sediment from bank erosion and upland delivery of forested soils? How are the carbon loads determined? Is this particulate carbon associated with sediment or something else? What about large wood and CPOM? What about the role of floods in reservoirs which can introduce substantial large wood and sediment? What about reservoir infilling? This is perhaps not an issue on the largest reservoirs but could be important to numerous mid-sized reservoirs over the course of the model timeframe. Here is just one example:

Zhang, Qian, Robert M. Hirsch, and William P. Ball. "Long-term changes in sediment and nutrient delivery from Conowingo dam to Chesapeake Bay: effects of reservoir sedimentation." *Environmental science & technology* 50, no. 4 (2016): 1877-1886..

I think many of these questions could be addressed with a concise presentation of the output of Global-NEWS within the methods or Supplementary Information. I also think a brief consideration of the scenarios within Global-NEWS should be provided to a reader who is not familiar.

Carbon fluxes associated with geomorphic processes, at least those with a recurring probability of a year or less, are accounted for in the Global-NEWS model. POC and particulate nutrients are related to the total suspended sediment (TSS) concentrations, which in turn are predicted taking into account the controlling factors of soil erosion: slope, land use, climate (temperature and precipitation) and lithology (Beusen et al., 2005). These factors also control stream bank erosion, in particular because the average slope of the river catchment correlates with the slope of the stream channel, and because climate variables are related to stream flow and density of the river network. Stream bank erosion contributes to observed TSS fluxes on which the empirical Global-NEWS model is trained. Thus, stream bank erosion is implicitly included in the estimated POC and nutrient loads. Note further that Global-NEWS predicts annual loads at the watershed scale and therefore does not resolve local effects of flooding on river hydrology (Fekete et al., 2010).

A limitation of Global-NEWS is the lack of representation of extreme hydrological events with a low recurring probability but potentially large contributions to long-term material

fluxes. These events are not captured, neither by the model nor by the observed data it was trained on. These events can involve geomorphologic processes such as landslides, which mobilize the carbon stored in the entire soil horizon plus the standing biomass. Such events, which can be related to extreme weather phenomena (e.g. tropical cyclones) or earthquakes, have been described for steep catchments in the sub-tropics (e.g. Taiwan, West et al. 2011) and temperate regions (e.g. New Zealand, Hilton et al. 2011). Their contribution to POC mobilization at the global scale still needs to be quantified.

The reviewer brings up additional pertinent points regarding the delivery of large wood debris and reservoir infilling, which may change the water residence time and sedimentation rates as reservoirs age. Vorosmarty et al. (1997) estimated that sedimentation reduced the volume of reservoirs in the United States by up to 2 km³, or ~0.2% of the total US reservoir volume. To our knowledge, there does not exist a more up-to-date estimate of this value, nor is there a global estimate. In our previous work on the retention of P and Si in reservoirs, we extensively searched for data on the dredging of reservoir systems, which increases reservoir volume. These data, however, are extremely difficult to obtain, even in our home countries. We therefore acknowledge that reservoir infilling and dredging introduce uncertainty in our model estimations. In light of Vorosmarty et al. estimate, we believe this uncertainty is likely small relative to the other sources of uncertainty in our modeling approach (and discussed in our submission).

To address the reviewer's points, we have expanded our description of the Global-NEWS yield estimates in the third paragraph of the Methods section as follows.

"The Global-NEWS model is well suited for the proposed modelling approach: it differentiates between POC and DOC, it implicitly accounts for in-stream OC losses, and it has been used to hindcast nutrient loads to watersheds in the years 1970 and 2000, and to forecast the 2030 and 2050 loads according to the Millennium Ecosystem Assessment (MA) scenarios. Global-NEWS predicts sediment, OC and nutrient yields based on land use and land cover (e.g., wetlands, cropland, and grasslands), climate variables (including temperature and precipitation), geomorphological parameters (including slope and lithology), and anthropogenic alterations (including consumptive water usage). The MA scenarios are storylines for a future world that will become either more globalized (Global Orchestration, GO, and TechnoGarden, TG) or regionalized (Adapting Mosaic, AM, and Order from Strength, OS), and take either a proactive (TG and AM) or reactive approach to environmental management (GO and OS)⁵. Each of the MA scenarios assumes changing land uses, climate regimes, and anthropogenic perturbations, which in turn modify the fluxes of sediments, OC and nutrients delivered to river systems.

A caveat of Global-NEWS is the lack of representation of extreme hydrological events with a low recurring probability but potentially large contributions to long-term riverine fluxes. These events are not captured by the Global-NEWS model itself or by the observed data it was trained on. These events can involve geomorphologic processes such as landslides, which mobilize the carbon stored in the entire soil horizon plus the standing biomass. Such events, which can be related to extreme weather phenomena (e.g. tropical cyclones) or earthquakes, have been described for steep catchments in the sub-tropics (e.g. Taiwan (West et al. 2011) and temperate regions (e.g. New Zealand, (Hilton et al. 2011)). Their contribution to OC mobilization at the global scale remains to be quantified.

In our modeling approach, reservoir infilling due to sedimentation is not taken into account. Vorosmarty et al. (1997) estimated that sedimentation has reduced the volume of reservoirs in the United States by up to 2 km³, that is, only ~0.2% of the total US reservoir volume. While reservoir infilling may vary significantly from one reservoir to another, the effect of sediment accumulation on water residence time likely represents a relatively minor source of uncertainty on the impact of dams on the global riverine OC flux.”

Line 29 – are all references to OC assumed to be TOC?

Yes. We have clarified this in the abstract and introduction.

Line 29 to 31 – these two sentences are confusing as they are written.

We have rephrased these sentences so they now read:

“Humans, however, have profoundly altered the balance between carbon fixation, mineralization and OC burial along the river continuum, not only by increasing the loadings of OC and nutrients to rivers but also through the massive building of dams⁴⁻⁷ .

Line 44 – specify within the reservoir region.

Changed.

Lines 87 to 89 – This is not entirely clear as it is written.

We have rephrased this section so it reads:

“For any given reservoir, the input fluxes of allochthonous POC and DOC supplied by the upstream catchment are obtained from the watershed yields estimated by the Global-NEWS model. The yields account for the effects of climate change (temperature and hydrology) as well as land use and land cover changes on POC and DOC loadings to rivers^{20,31}”

Line 93 – this needs a bit of clarifying. What comes out via dam outflow? Are you assuming 100% trapping of POC and release of DOC? Figure 1 is not completely clear and refers to TOC coming out downstream. How do you know what is released from one reservoir and into the next? Or is this process not considered explicitly and you just account for the input downstream from the next set of Global-NEWS output (according to Figure S2).

In Figure 1, the global outflows of DOC and POC from all reservoirs at the stated time point are added together for both allochthonous and autochthonous OC. In other words, the TOC outflows comprise both DOC and POC (we do not assume 100% trapping of POC). Similarly, we have combined the DOC and POC inflows and present them as a single TOC inflow. (Note however, that only POC is assumed to be buried in reservoir sediments, which is entirely reasonable.) We have combined the DOC and POC inflows and outflows in order to keep the number of fluxes (arrows) in the figure manageable. As such the figure illustrates the impact of dams on the riverine fluxes of total OC. Table S6 in the supplementary material provides the breakdown of the TOC fluxes into their DOC

and POC components. In the revised caption of Figure 1, the TOC inflows and outflows are clearly identified as follows:

“ TOC_{in} : global influx of POC plus DOC to dam reservoirs (note: the routing procedure avoids double counting OC passing through cascades of dams, see Figure S2). P : primary production. TOC_{out} : global efflux of POC plus DOC exiting dam reservoirs, without double counting TOC that passes through multiple dams.”

The allochthonous POC and DOC fluxes leaving a reservoir are added to the POC and DOC loadings from the landscape to the river below the dam (as estimated using Global-NEWS). The combined fluxes are then the POC and DOC inflows to the next downstream reservoir (if there is one). This explained in detail in the Methods section when introducing equation (10).

Line 126 - why does the HRT of reservoirs decrease after this time? Are they smaller or built differently to reduce the HRT?

The majority of large dams now being built, or projected to be completed by 2030, are hydroelectric dams (Zarfl et al., 2016), in contrast to the dams built before the turn of the century, which were often designed to provide water storage capacity and flood control. On average, hydroelectric reservoirs tend to have lower water residence times than water storage reservoirs because their main priority is to pass a large volume of water through the turbines, resulting in high discharge. In contrast, storage reservoirs release little water through the dam for much of the year and thus exhibit higher hydraulic residence times. We have clarified this on lines 126-128.

Line 183 – Is it possible that you are overlooking key processes of delivery? This is difficult to assess in the current version. Caveats of the modeling approach should be presented somewhere.

Our model relies on the Global-NEWS inputs of OC to watersheds, which we now describe in more detail in response to the reviewer’s first comment. Caveats of the modelling approach, including the uncertainty associated with relying on the Global-NEWS model for reservoir inputs, are presented in detail in the sensitivity analysis and uncertainty sections, which we have moved from the Supplementary Material to the Methods section. In particular, we point to the fact that extreme events including major flooding and landslides may play an important role for long-term OC fluxes, but they are not included in GlobalNEWS and they are still poorly constrained at the global scale.

Figure 1. Cascades of dams – I am having trouble interpreting the cascades of dams. This assumes that DOC is passing through a reservoir, but that POC is not, correct?

No. See our response to the comment concerning Figure 1 above: both DOC and POC are exported through dams. For the allochthonous OC this is done explicitly, while for autochthonous OC, DOC and POC are combined because of the difficulty of parameterizing the transformations between the two pools of fresh, in-reservoir produced OC.

Based on the schematic, I think the reader might get the incorrect assumption that the output is something coming from out of a dam rather than the budget for all reservoirs globally. It looks like a single reservoir pictured in the background and that might be

misleading. This isn't really the output from a dam, but what reaches the ocean from all dams, right?

We have revised the figure caption and clearly state that the fluxes shown correspond to the aggregated values for all reservoirs globally. The effluxes represent the total fluxes that leave dams worldwide, without double-counting OC that passes through multiple cascading dams (which is stated in the caption). Note that the total OC exported from reservoirs is not equal to the OC that reaches the ocean, which is a larger flux because it includes the fraction of the OC loading to rivers that never passes through a dam.

Are there any consequences for the carbon if it passes through multiple dams?

Carbon that passes through multiple dams has greater opportunity to be transformed, mineralized, or buried. This process of increased processing and increased retention of carbon due to cascading dams is explicitly simulated in our model (see Figure S2).

Supplementary Information

Figure S1 – I think it might be more useful for this figure to be more process based if possible, showing a water column flowing into a reservoir and linking various processes through arrows. Or consider a budget approach where inputs are shown as positive contributions and outputs are considered negative. The caption needs clarity. What are the different arrow colors representing?

We appreciate the suggestions of the reviewer and have considered the two options. However, in the end, we prefer to retain the simple box model representation shown, because it most accurately reflects the (simple) modeling approach used to link inputs, transformation processes and outputs. Attempting to visualize in more detail the hydrology and biogeochemistry of a reservoir system would possibly suggest that we are modelling processes at a higher spatial and temporal resolution than is actually the case, or that we use a particular type of dam reservoir as a template for all reservoirs worldwide. Further note that the OC solubilisation and mineralization processes take place in the water column, riparian zones and the upper part of bottom sediments. The colours don't mean anything: we have changed the figure to grey-scale.

Figure S2 – caption needs clarity. K1 a model node? A dam? Although it is defined in the text, it should be included in the caption.

We have revised the caption so that it now reads:

“Schematic representation of the breakdown of a hypothetical watershed into the sub-watersheds that are hydrologically connected to the dam reservoirs in the watershed; k represents the most downstream dam, $k-1$ the next dam upstream, and so on. The corresponding sub-watershed for dam k is W_k , W_{k-1} for dam $k-1$, and so on. The figure helps explain the routing procedure described in Methods section 4 and equation 10.”

Figure S7 caption – Line 245 – this should be outliers, correct? See comment regarding discussion of this figure in the text.

Yes, these are outliers. The revised caption of Figure S7 now reads:

“Comparison between k_{min} values generated by the Monte Carlo (MC) procedure used in our model and the k_{min} values obtained independently by Catalan et al.³⁰ from a global data compilation. The first three boxes to the left show the output of the MC analysis with the default model constraints (Table S2). The boxes labelled “Auto, x3”, “Auto, x1”, and “Auto, x6” show additional outputs of MC analyses where the reactivity of autochthonous OC is assumed to be 3 times higher, equal, and 6 times higher than that of allochthonous OC, respectively. The last box shows the k_{min} distribution of Catalan et al., which lumps together values for POC and DOC, and for allochthonous and autochthonous OC. For clarity, extreme outliers of the Catalan data are not shown. Note that the default scaling factor of 3 used in the OC reservoir model (i.e., imposing a mean k_{min} value 3 times higher for autochthonous than allochthonous OC) is consistent with the observed k_{min} distribution of Catalan et al., while this is not the case for the lower (equal reactivity) or higher (6 times higher) scaling factors.”

Methods

Line 24 – how is success established? What is the criteria for successful prediction?

We realize that the formulation of this sentence is unfortunate as it may suggest that the model predictions were directly validated against existing global databases for Si and P retention. We have therefore removed the word “successfully.”

Line 89 – Why? These are riverine fluxes – I understand you are dealing with reservoirs that might mimic lentic processes, but do you have evidence to suggest this is similar for riverine fluxes?

Yes. The available evidence suggests that worldwide patterns of DOC concentrations are similar between lakes and rivers. The Sobek estimate we used predicts a mean DOC concentration of 7.2 ppm, with values distributed according to a gamma PDF, with values rarely above 40 ppm. Existing global estimates of riverine DOC concentrations are in close agreement with these values; Meybeck (1988) estimates a mean global riverine DOC concentration of 6.3 ppm, while estimates of DOC concentrations in the world’s 20 largest rivers also yield mean values on the order of 6-7 ppm (Spitzky and Leenheer, 1991). However, the river DOC estimates are based on much smaller datasets than Sobek’s 7000 lakes, and thus deriving a global PDF solely on existing river data is questionable. We have revised the text to include the river DOC references mentioned above, and we added a brief justification for using Sobek’s lake distribution.

Line 129 – This figure does not show the outliers. The way it is referred to suggests that it might.

We have clarified the text and the figure caption. The revised caption of Figure S7 is given above. The text in the methods section now reads:

“To capture the range of the Catalan, et al.²⁵ rate constant data, with the exception of 4 outliers, 3 of which are from the same small streams, the autochthonous scaling factor is assumed to follow a normal distribution between 1 and 6 (Table S1). With this range of the scaling factor, the model-generated total OC mineralization rates reproduce the range of observed rate constants of Catalan, et al.²⁵ (Figure S7).”

Line 179 to Line 181 – I think this needs to be more specific, especially with respect to sedimentation rates. Additionally, some of the references are dated.

We have expanded the description of the burial rate constant as follows:

“The burial rate constant, k_{bur} , is an effective parameter describing the long-term retention of POC with the sediments accumulating in the reservoir, that is, the POC that is not remineralized or otherwise remobilized and exported over reservoir’s lifetime. The value of k_{bur} aggregates all the factors controlling the POC burial efficiency other than the water residence time, including for instance sedimentation rate, reservoir morphology (which controls deposition patterns), oxygen exposure time, temperature, and sediment resuspension events¹⁸.”

We agree that some of the references are dated, but that does not disqualify them from inclusion in our analysis. In fact, many of the standard studies on organic matter degradation and preservation in active depositional environments were carried out during the second half of the 20th century. We therefore opted to consider all the reliable data and aimed for the broadest spatial distribution of studies, in order to derive a global distribution of burial rate constants. This resulted in a combination of older references and newer references (Kastowski et al., 2011; Sobek et al., 2009). We have added a statement urging for more studies quantifying burial rate constants in a wide diversity of reservoir settings.

Line 215 – Is there any way to provide additional information about these scenarios for a reader who is unfamiliar?

We provide a brief description of these scenarios in the Methods section, lines 47-53.

Line 229 – cascading dams change to dams that occur longitudinally in series to be more specific if possible.

Changed.

Skalak, Katherine J., Adam J. Benthem, Edward R. Schenk, Cliff R. Hupp, Joel M. Galloway, Rochelle A. Nustad, and Gregg J. Wiche. "Large dams and alluvial rivers in the Anthropocene: The impacts of the Garrison and Oahe Dams on the Upper Missouri River." *Anthropocene* 2 (2013): 51-64.

We thank the reviewer for this reference and have added it when we introduce sediment trapping and burial in dams (lines 39-41).

Line 247 – an initial hypothesis...

Changed.

NOTE: In addition to the above changes, we have moved the description of the uncertainty and sensitivity analysis from the supplementary material into the methods section at the suggestion of the editor.

REVIEWERS' COMMENTS:

Reviewer #3 (Remarks to the Author):

The authors have addressed all comments to my original review completely and to my satisfaction. In the case where the authors did not change the paper according to a suggestion I provided (in the case of Figure S1 as an example), they provided sufficient justification for maintaining their original approach and provided adequate supporting language in the manuscript to reduce any confusion that had arisen from the original version.

I have no additional comments or suggestions and think that the manuscript is acceptable in its current form for publication in Nature Communications.

The addition discussion on Global NEWS in particular provides more clarity for a reader not completely familiar with the approach and greatly facilitates interpretation of results and significance. The editor's suggestion of including the uncertainty and sensitivity was also a helpful improvement.

I think this contribution will provide a helpful advancement for those working in global carbon cycles, but also in particular for those considering fluvial and reservoir processes within the contexts of dams and also dam removal.

REVIEWERS' COMMENTS:

Reviewer #3 (Remarks to the Author):

The authors have addressed all comments to my original review completely and to my satisfaction. In the case where the authors did not change the paper according to a suggestion I provided (in the case of Figure S1 as an example), they provided sufficient justification for maintaining their original approach and provided adequate supporting language in the manuscript to reduce any confusion that had arisen from the original version.

I have no additional comments or suggestions and think that the manuscript is acceptable in its current form for publication in Nature Communications.

The addition discussion on Global NEWS in particular provides more clarity for a reader not completely familiar with the approach and greatly facilitates interpretation of results and significance. The editor's suggestion of including the uncertainty and sensitivity was also a helpful improvement.

I think this contribution will provide a helpful advancement for those working in global carbon cycles, but also in particular for those considering fluvial and reservoir processes within the contexts of dams and also dam removal.

We thank reviewer 3 for taking the time to read through our manuscript a second time, and are grateful for the feedback she provided, which has improved our manuscript. Given that she has requested no additional changes during this round of review, we have not altered the manuscript in any way other than as requested by the editor (see cover letter file detailing changes).